

# A guide to optimised spatiotemporal data co-location by mutual information maximisation

Andrew Steven Martin[1,2], Heather Guy[1,2], Michael Ray Gallagher[3,4], and Ryan Reynolds Neely III[1,2]

[1]School of Earth and Environment, University of Leeds, Leeds, U.K.
[2]National Centre for Atmospheric Science, Leeds, U.K.
[3]Cooperative Institute for Research in Environmental Sciences, University of Colorado, Boulder, Colorado, USA
[4]NOAA Physical Sciences Laboratory, Boulder, Colorado, USA

**Correspondence:** Andrew Steven Martin (eeasm@leeds.ac.uk)

**Abstract.** The matching of data described on different coordinate systems between multiple data sources – spatiotemporal co-location – is a necessary and crucial step in geospatial data synthesis and validation. The particular choice of co-location scheme, and the choice of parameters applied to it, decide what subsets of the original datasets are included in downstream analyses, affecting the quantitative outputs of comparison studies and multi-retrieval synthesised datasets. Previously, no gen-

eralised framework for deciding how best to co-locate data has existed. We outline a domain- and data-agnostic framework that generalises the process of selecting an optimised co-location parametrisation for a given co-location scheme, by maximising the mutual information encoded between the data included in the subsequent analyses. We demonstrate the framework by applying it to a comparison of vertical cloud fraction profiles retrieved from the polar-orbiting ICESat-2 satellite's ATL09 data product, and surface-based observations at four Cloudnet observatories. We evaluate per-site optimised co-location parametrisations and

find that using the optimised co-location parametrisations quantitatively improves the comparison between the datasets over naive choices of co-location parameters. This work has implications across almost all remote sensing data products – especially for satellite validations – and will facilitate deep learning methodologies by producing paired datasets with the maximal information about the structure between datasets available to be learned.

## 1 Introduction

Remote sensing data, obtained from Earth observation satellites and surface based observatories, provide invaluable data for furthering our understanding of Earth-system processes, for the validation an constraining of models, and for making observations in remote locations or locations with extreme conditions.

Particularly for satellite data, rigorous validation and a formal uncertainty characterisation are essential for subsequent use of the data. In order to validate satellite data, we need to compare it against reference measurements (Loew et al., 2017). Rarely

are the reference measurements described on the same set of coordinates as the data to be validated. Inter-comparison of remote sensing retrievals and multi-sensor data synthesis are subject to similar challenges.

Ideally when comparing data from two different sources, the observations are made simultaneously, and are sensitive to the same spatial volume. However, observations from different platforms will have different viewing geometries, such that





the sensitivities across the same observed spatial volume differ between the data sources. This induces a difference between
the measurements often referred to as the smoothing error (e.g. Rodgers, 2008). Furthermore, the observations from different
sources can measure distinct physical volumes that are spatiotemporally displaced from each other. This induces a bias com-
monly referred to as the co-location mismatch (e.g. Verhoelst et al., 2015; Virtanen et al., 2018). Verhoelst et al. (2015, Fig. 1)
and Loew et al. (2017, Fig. 2) both provide good representations of the issue of co-locating measurements.

In order to compare data recorded on different sets of coordinates, we need to perform spatiotemporal co-location. We define
spatiotemporal co-location as the process of matching data between two or more data sources, described on different sets of
coordinates, such that discrete co-location events can be defined. For a given co-location event, the data associated with it
from the different data sources is considered sufficiently close in time and space to be directly comparable once the data have
been homogenised (Loew et al., 2017). Often, for a given implementation of spatiotemporal co-location – which we will refer
to as a *co-location scheme* (defined in Sect. 2.1) – there will be parameters for the scheme that change the amount of data
permitted by the subsetting operations of the co-location process. Once data have been co-located and co-location events have
been identified, formal uncertainty characterisation can be performed, or other comparison metrics such as the bias, RMSE and
correlation coefficients can be calculated (Loew et al., 2017, Sect. 3.4).

A good spatiotemporal co-location requires that there are sufficient co-location events for the subsequent analysis to be
viable. Conversely, the co-location cannot permit so much data that the subsequent analysis is contaminated with data being
compared between two physically independent sets of observations. Finding a parametrisation for a co-location scheme that
balances the need for sufficient data, whilst minimising the co-location mismatch between the compared data within a co-
location event is the crux of the problem.

As an example, when comparing data between a satellite and a surface based observatory, a simple and often used co-location
scheme is to generate co-location events when the satellite measurement footprint falls within some along-ground distance $R$
of the surface based observatory, and to subset the surface based data with a temporal window of duration $\tau$, centred on the
time of closest approach of the satellite to the observatory. By doing this, individual co-location events (often described as
*overpasses*) are small segments of a single orbit from the satellite, and the data is often averaged along-track to obtain a single
vertical profile or scalar value that can be compared against temporally averaged data from the surface-based observatory (e.g.
Alexander and Protat, 2018; Baars et al., 2023; Liu et al., 2017; Lu et al., 2021; Mamouri et al., 2009; Martin et al., 2021;
Mona et al., 2009; Pappalardo et al., 2010; Pauly et al., 2019; Proestakis et al., 2019; Protat et al., 2009; Robinson et al.,
2025; Schuster et al., 2012). Each of the studies using the aforementioned co-location scheme to match data between a satellite
and surface-based observatory need to make a choice for the values of $R$ and $\tau$, the parameters affecting the spatiotemporal
volumes within which data must have been recorded in order to be permitted in subsequent analyses.

When deciding how data are co-located, care must be taken to ensure the co-location parametrisation is selected indepen-
dently of the results of the subsequent analysis (von Clarmann, 2006). Some studies justify the choice of their co-location
parametrisations qualitatively (e.g. Baars et al., 2023; Blanchard et al., 2014; Fuchs et al., 2022; Lu et al., 2021; McErlich
et al., 2021; Proestakis et al., 2019; Robinson et al., 2025; Sayer et al., 2020). Some studies do test the effects of changing the
co-location parameters empirically (e.g. Alexander and Protat, 2018; Eibedingil et al., 2021; Pappalardo et al., 2010; Protat





et al., 2009) however, these studies use the comparison metric being used in the subsequent analysis to justify or inform the
choice of co-location parametrisation. This should be avoided, as the comparison metric is an unknown quantity (hence the
need for the analysis in the first place), and by using the value of the comparison metric to inform the co-location parametrisa-
tion – which in turn affects the estimate of the comparison metric itself – we effectively apply a prior expectation to the to the
comparison metric in the analysis.

For example, the linear correlation coefficient between retrieved values is often used as a comparison and validation met-
ric. Often, the co-location parametrisation that maximises the correlation coefficient is selected (e.g. Eibedingil et al., 2021;
Pappalardo et al., 2010; Schuster et al., 2012). The computation of the correlation coefficient is an estimate, with bias and
variance depending on the data permitted by the co-location. By selecting a co-location parametrisation that maximises the
correlation coefficient, we are preferentially selecting results that are biased high. This is akin to over-fitting models to our
data. The outcome: results will look better than they potentially are. For example, the uncertainty budgets for retrievals may be
underestimated and inferred biases could be too small in magnitude.

As such, an independent metric for assessing the quality of data co-location is necessary. By treating measurements as sam-
ples drawn from probability distributions of underlying geophysical fields, and paired measurements within a co-location event
as being drawn from a joint probability distribution, we propose the mutual information between data within co-location events
as an appropriate metric to assess the quality of spatiotemporal co-locations. Mutual information balances the requirements for
sufficiently sampling the available system states such that we can infer relationships between the data, whilst being sensitive
to the inclusion of comparisons between physically independent samples.

From this, we outline a generalised framework for evaluating optimised co-location parametrisations by maximising the
mutual information between data to be compared. For any co-location scheme that can be parametrised with a finite number
of parameters (Sect. 2.1), the mutual information (Sect. 2.2) is estimated between the data permitted by the co-location (Sect.
2.3), and the optimised co-location parametrisation is selected as the parametrisation that maximises the mutual information
(Sect. 2.4). We demonstrate the framework by applying it to a comparison between the ICESat-2 ATL09 cloud layer product
and macrophysical cloud products derived from four Cloudnet observatories (Sect. 3).

## 2   Framework

### 2.1   Framework definitions

The outcome of this framework is the co-location of data where the information shared between the two retrievals is max-
imised. As described in Loew et al. (2017), before validation metrics between datasets can be calculated, three key steps must
be performed: quality checks, to ensure the data being compared are realistic, self-consistent, and reasonably well charac-
terised; spatiotemporal co-location, ensuring that the data being compared represent sufficiently similar measurements in time
and space, and; homogenisation, whereby any further transformations to the data (unit conversions, temporal or spatial aggre-
gations, etc) are applied, allowing like-to-like comparisons to be made between the homogenised data. In this framing, both the
quality checks and co-location processes act to subset the data, permitting observations that meet both the quality requirements



and co-location criteria to be used in the homogenisation process. Throughout this work, we will refer to co-location schemes, criteria, parametrisations and events.

*Co-location scheme*: the method by which data from two or more data sources are matched with each other. Some example
co-location schemes are outlined in Fig. 1.

*Co-location criteria*: the logical statements that implement a given co-location scheme. The co-location criteria often take the form of inequalities, with data satisfying the inequalities being included in the analysis.

*Co-location parametrisation*: a vector specifying the values of variable parameters used in the co-location criteria. These can be described by a general parametrisation vector, $\boldsymbol{p} = (p_1, \ldots, p_M)$, where $M$ is the number of values required to fully
describe the co-location scheme.

*Co-location event*: a discrete unit of matched homogenised data between the co-located data sources that simultaneously satisfy all of the co-location criteria.

The framework requires that the co-location criteria for a given co-location scheme can be be described by a finite number of parameters, which is applicable for all realistic co-location schemes. The co-location scheme *can* be arbitrary, but it should
be physically motivated to achieve the best results. Figure 1 shows four possible co-location schemes between data of different dimensionalities. Panel (a) shows a scheme for co-locating satellite swath data and point-like surface based observations, as was described in the introduction (e.g. Alexander and Protat, 2018; Baars et al., 2023; Blanchard et al., 2014; Liu et al., 2017, 2010; Lu et al., 2021; McErlich et al., 2021; Pappalardo et al., 2010; Proestakis et al., 2019; Protat et al., 2009; Robinson et al., 2025; Schuster et al., 2012). Panel (b) instead shows a possible scheme for matching data between a 2-dimensional source (e.g. a grid
of pixels) and a point source of data (e.g. Compernolle et al., 2021; Deneke et al., 2009; Eibedingil et al., 2021; Fuchs et al., 2022; Papagiannopoulos et al., 2016; Protat et al., 2014b; Roebeling et al., 2008; Tzallas et al., 2019; Vinjamuri et al., 2023). Figure 1c shows a possible co-location scheme between two 1-dimensional data sources – possibly two satellite ground tracks (e.g. Protat et al., 2009; Wang et al., 2024). Data are subset based on it falling within a circle of radius $R$, common to both data sources, and centred on the location where the paths cross. There is a second criterion, that the time difference between
the data sources being at the crossing point should be less than some upper bound $\tau$, where $r_i$ are the distances of data from source $i$ to the crossing point, and $t_i$ are the times associated with data from source $i$ being at the crossing point. Thus, the co-location scheme leads to the co-location criteria of $r_i \leq R$ and $|t_1 - t_2| \leq \tau$. The co-location criteria can be described by the parametrisation $\boldsymbol{p} = (R, \tau)$. co-location events consist of paired homogenised data between the two satellites where their orbital paths intersected within a time $\tau$, and the data are spatially subset within the circle of radius $R$.

Although the above co-location schemes are basic and naive to the underlying physical processes that govern the spatiotemporal gradients of the measurands, our knowledge of the underlying processes can be encoded into the co-location scheme through the inclusion of additional co-location criteria and higher dimensional co-location parametrisations. For example, more complex schemes could allow us to encode our expectations of how local advection would affect the spatial distribution of (in)dependent samples between data sources, through the implementation of logical criteria on ancillary wind data.





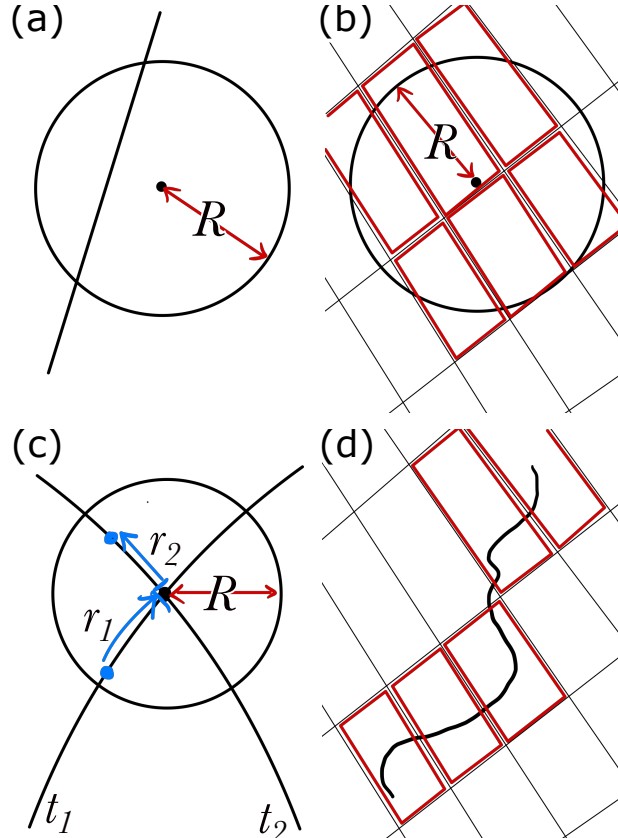

**Figure 1.** Example realisations of spatial co-location schemes between data of different spatial dimensionalities. **(a)** A point-line co-location, where the data falling within a distance $R$ of the point data source is utilised from the line source. **(b)** A point-area co-location, where pixels whose centres fall within a distance $R$ of the point data source are used. **(c)** A line-line co-location where data falling within a distance $R$ of the crossing point between the lines is used. **(d)** A line-area co-location, where a minimum path length, $l$, must be traced within each pixel in order for the pixel to be used in the analysis. In **(b)** and **(d)**, the pixels highlighted in bold (red) are those selected to remain in the homogenisation process. Each spatial co-location scheme will also be paired with a temporal co-location scheme.

## 2.2 Mutual information

In this framework, we treat the underlying physical state as being independent between co-location events. Thus, we treat the underlying physical state of the system being measured as a random variable drawn from the distribution of all plausible system states. Measurements are affected by this randomness, as well as other confounding variability due to co-location mismatch and detector noise (for example). As such, pairs of measurements within a co-location event should be related by a joint probability distribution that accounts for the distribution of system states and the additional variability. If the measurements being made are not independent, the joint probability distribution will have some non-independent structure that can be used to inform





us about the relationship between the measurements. Mutual information is a concept derived from information theory (e.g. Shannon, 1948) that we use as a quantitative metric to assess the quality of the relationship between sets of retrievals when co-locating the data with different co-location parametrisations.

Nearing et al. (2017) describes the entropy of a variable $X$, $\mathrm{H}(X)$, as a measure of our ignorance about the variable $X$ prior to observation, and as the average quantity of information we stand to learn by measuring $X$.

$$\mathrm{H}(X) = \mathbb{E}_X \left[ \log \left( \frac{1}{p(X)} \right) \right], \tag{1}$$

where $p(X)$ is the probability of the variable having a given value, and $\mathbb{E}_X$ is an expectation value taken over all possible values of $X$. Nearing et al. (2017) describes mutual information between two random variables $X$ and $Y$ as the expected reduction in

our ignorance of the possible values of $X$, given our knowledge of the value of $Y$. Mutual information is expressed as

$$\mathrm{I}(X;Y) = \mathbb{E}_{X,Y} \left[ \log \left( \frac{p(X,Y)}{p(X)p(Y)} \right) \right], \tag{2}$$

where $p(X,Y)$ is the joint probability of the outcome of $X$ and $Y$ occurring simultaneously, $p(X)$ and $p(Y)$ are the marginal distributions of $X$ and $Y$ respectively, and $\mathbb{E}_{X,Y}$ represents an expectation over all possible pairs of $X$ and $Y$. Mutual information has units depending on the base $b$ of the logarithm used in the information theoretic equations, with $b = 2$ giving

units of $\mathrm{bits}$, and $b = e$ giving units of $\mathrm{nats}$. Conversions between information theoretic units consists of linearly scaling the information theoretic values.

In our framework, $X$ and $Y$ are the retrievals or measurements of underlying physical quantities that we wish to compare or relate. In the limiting case that $X$ and $Y$ are independent, we obtain $p(X,Y) = p(X)p(Y)$, and Eq. (2) yields a value of $\mathrm{I} = 0$. Otherwise, the mutual information encoded between the measurements will be positive, with larger values indicating greater

reductions in our ignorance of the pair of measured values given access to one of the values.

When co-locating data between data sources in order to compare the data, we want to include data that best characterises the relationship between the data sources. The best possible relationship between our data is a one-to-one mapping between values $X$ and $Y$. In this case, knowledge of one variable fully determines the value of the other. This case is equivalent to minimising our ignorance of the joint value of the two retrievals, and is equivalent to maximising the mutual information between the

retrievals.

This is shown in Fig. 2. Two variables, $X$ and $Y$, each with a fixed marginal distribution (Fig. 2a–c), take on two distinct joint probability distributions (Fig. 2d–e). Figure 2d shows $X$ and $Y$ as being independent variables. The probability density is distributed throughout the space, and the probability of $X$ given any value of $Y$ simply follows the marginal distribution $p(X)$. In this case, learning the value of $Y$ yields no new information about the possible value of $X$, and the mutual information

is estimated to be near zero. The value of $\hat{\mathrm{I}}_{\mathrm{KSG}} = -0.001$ $\mathrm{nats}$ can be negative as a result of it being empirically estimated (see Sect. 2.3). In Fig. 2e, $X$ and $Y$ have a strong non-linear dependency. If we know the value of $Y$, our ignorance about the possible values of $X$ decreases substantially, as $X$ almost certainly falls on the manifold mapping $Y$ to $X$. There is obvious structure in the joint probability distribution that can be learned, and as a result, the mutual information estimate $\hat{\mathrm{I}}_{\mathrm{KSG}} = 3.167$ $\mathrm{nats}$ is higher when $X$ and $Y$ are dependent compared to being independent.



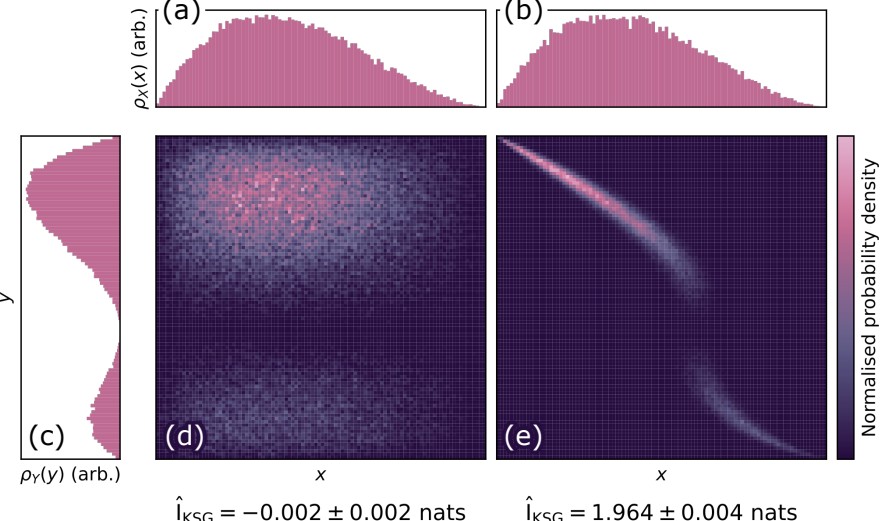

**Figure 2.** Two synthetically generated variables $X$ and $Y$ with fixed marginal distributions (**a**, **b** and **c**) form two different joint probability distributions (**d** and **e**). $X$ and $Y$ are independent in **d**, but have a strong non-linear dependence in **e**. When there is dependence between the variables, the probability density is spread across fewer possible states. Mutual information, $I_{KSG}$ (given below **d** and **e**), captures this structure, increasing as the individual variables encode more information about the joint distribution.

## 2.3 Mutual information estimation

The definition for mutual information given in Eq. (2) requires full knowledge of the joint probability distribution between the variables in question. We have incomplete knowledge of the marginal and joint probability distributions from which our measurements are sampled. Thus, we need to estimate mutual information, and the estimator must be able to handle a finite number of continuous valued samples as input. Problems in the Earth sciences may require comparison of multiple variables simultaneously, or of vector quantities. Thus, mutual estimators that also handle higher dimensional samples from probability distributions are preferable.

One method to estimate the mutual information is to discretise the measurements and produce a histogram approximating the underlying probability distributions (e.g. Beirlant et al., 1997), as is demonstrated in Fig. 2. This method requires that a sufficient number of joint measurements are made in order to well characterise the joint probability distribution. As mutual information is a measure of the structure of the joint probability distribution, the choice of histogram bins into which the data are discretised is very important. The bins need not be uniform in size, and adapting the bins to the data can decrease the bias and variance of the estimator (Darbellay and Vajda, 1999).

Another commonly employed method is estimating the mutual information from nearest neighbour distances between samples (e.g. Kraskov et al., 2004; Holmes and Nemenman, 2019). Regions of high probability density will likely be sampled more frequently than low probability density regions, resulting in samples drawn with low separations between them, indi-





cating structure in the distribution. Holmes and Nemenman (2019) describes a method that extends the mutual information estimators described in Kraskov et al. (2004), $\hat{I}_{\mathrm{KSG}}$, from 1-dimensional to multidimensional samples from both $X$ and $Y$, and a way to characterise the bias and variance of the estimator. The extension of a nearest neighbours method to multidimensional samples allows fast and (relatively) computationally efficient calculation of mutual information compared to discretisation
methods.

## 2.4 Parametrising co-location criteria by maximising mutual information

A good comparison between data requires that comparisons are made between retrievals from co-location events that well sample the physically possible underlying system states. This amounts to obtaining enough co-location events such that the marginal distributions of all retrievals are well sampled.

The reliable estimation of the mutual information requires a certain number of co-location events to be permitted by a co-location parametrisation $\boldsymbol{p}$. Some $\boldsymbol{p}$ will permit too little data for the mutual information estimators to learn structure in the joint probability distribution, resulting in an underestimation of the mutual information. At some point, parametrisations $\boldsymbol{p}$ will permit sufficient data for the mutual information estimator to accurately estimate $\mathrm{I}(X;Y)$ (assuming no contamination by independent data), when the individual measurement marginal distributions are well sampled. Thus, having parametrisations
permit more data leads to increases in $\hat{I}$, until the estimated $\hat{I} \approx \mathrm{I}$, and additional co-location events provide no new information about the joint distribution of the retrievals being compared.

At some point, parametrisations will produce co-location events matching data within a sufficiently large spatiotemporal volume, such that the data contributing to the co-location event from different sources originate from physically independent observations. This will contaminate the joint probability distribution being assessed with independent samples. Appendix A
demonstrates, using a toy model, that contaminating the comparison with independent data necessarily reduces the upper bound of the mutual information encoded between retrievals. Thus, for $\boldsymbol{p}$ permitting data co-location within large enough spatiotemporal volumes, increasing the spatiotemporal volume should act to decrease the mutual information.

Figure 3 implements the toy model described in Appendix A1 to demonstrate the effects of a co-location parametrisation permitting too little and too much data. Figure 3a shows a data limited regime, in which there are deficient samples for learning
the structure of the relationship between the variables reliably. The mutual information between the variables can be improved by further sampling, as is shown in Fig. 3b. In this case, there is no contamination and the significantly denser sampling results in a value of $\hat{I}_{\mathrm{KSG}} = 1.968 \pm 0.015$ nats. Figure 3(c–d) show the effects of contamination with independent data, and how it rapidly reduces the estimated $\hat{I}_{\mathrm{KSG}}$ values to near zero when the signal to noise ratio is highest in panel (d).

Thus, there are two main factors influencing $\hat{I}$: a data limited regime in which the inclusion of more data acts to increase $\hat{I}$
and; a contaminated data regime in which the inclusion of more data increases the proportion of independent samples being compared, which acts to reduce $\hat{I}$. We postulate that a parametrisation $\hat{\boldsymbol{p}}$ exists for which the mutual information is maximised, where the competing effects of including more comparable samples and more independent samples are balanced. At $\hat{\boldsymbol{p}}$, the relationship between the retrievals, encoded in their joint distribution, is best characterised, and this should be the co-location parametrisation used in any subsequent analysis.



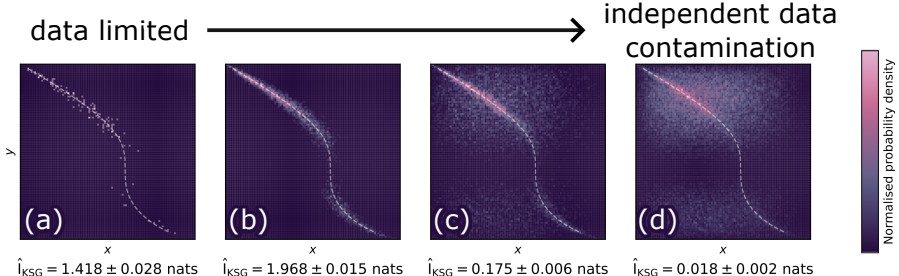

**Figure 3.** A demonstration of how a data-limited **(a)** and independent data contaminated **(c-d)** regime reduces the estimated mutual information encoded between two variables $X$ and $Y$ when compared to a case when $X$ and $Y$ are well sampled without contamination from independent data **(b)**. In all cases, $X$ and $Y$ are synthetic and generated from the same marginal distributions as in Fig. 2.

Thus, in order to optimise the co-location parametrisation $\hat{\boldsymbol{p}}$, the steps are:

1. Identify a set $\mathcal{P} = \{\boldsymbol{p}\}$, describing a range of plausible parametrisations.

2. For every $\boldsymbol{p} \in \mathcal{P}$, perform quality checks and spatiotemporal co-location subsetting according to $\boldsymbol{p}$ on the data to be compared.

3. Apply the chosen mutual information estimator to the co-located homogenised data, to obtain $\hat{I}(\boldsymbol{p})$.

4. Identify the optimised co-location parametrisation $\hat{\boldsymbol{p}}$ as the parametrisation maximising $\hat{I}(\boldsymbol{p})$. That is

$$\hat{\boldsymbol{p}} = \underset{\boldsymbol{p} \in \mathcal{P}}{\arg\max} \left( \hat{I}(\boldsymbol{p}) \right). \tag{3}$$

In the following Section, we will demonstrate the application and usefulness of this framework by co-locating satellite and surface based retrievals of vertical cloud fraction, and showing that the comparison is optimised by choosing the co-location parametrisation $\hat{\boldsymbol{p}}$ over other values.

## 3 Application example: validating the ICESat-2 ATL09 cloud layer product using Cloudnet observations

### 3.1 ICESat-2 ATL09 cloud layer product

ICESat-2 is a polar orbiting satellite, launched by NASA in 2018, and is the only satellite currently in orbit with the capability to make vertically resolved observations polewards of $83°$ north and south. The satellite has a single instrument payload, the Advanced Topographical Laser Altimeter System (ATLAS) – a photon counting lidar predominantly designed for altimetry (Neumann et al., 2019). To aid analyses of the altimetry data, atmospheric backscatter products are produced to facilitate quality checks on the altimetry data products. The ATL04 normalised relative backscatter profiles product is a level 2 product derived from the photon point-cloud data provided by ATL02 (McGarry et al., 2021; Palm et al., 2021a). ATL04 consists of



photon returns summed over $400$ consecutive laser pulses, giving a data product with $30$ m vertical resolution and $240$ m along-track resolution. Photon counts are reported in a vertical range from $250$ m below an on-board digital elevation model

(DEM) to $13.75$ km above the DEM. The ATLAS lidar transmits six laser beams, split into three pairs of a strong and weak beam, with the strong beams transmitting four times more power than the weak beams. The ATL04, and subsequently the ATL09 product, use the measurements from the three strong beams, producing three sets of vertically resolved observations.

The ATL09 calibrated backscatter and atmospheric layers data product (Palm et al., 2021b, 2023) derives from the ATL04 data product, with the aim of characterising the state of the atmosphere through which ICESat-2 performs its altimetry mea-

surements. Due to the challenges associated with absolute calibration of the backscatter profiles, the high noise rate, and the folding of signals into a $15$ km window (Palm et al., 2021b, a), a bespoke cloud detection algorithm was developed, the density dimension algorithm (DDA, Herzfeld et al., 2021a, b).

Although ATL09 calibrated backscatter profiles have been compared against profiles from a cloud physics lidar, and CALIPSO (Palm et al., 2021b), and the DDA has been demonstrated for cloud and blowing snow detection over Antarctica (Herzfeld et al.,

2021a), no validation of the produced cloud layer product has been made against surface based cloud observations. This study will demonstrate the framework outlined in Sect. 2 whilst providing an initial comparison of the ICESat-2 ATL09 cloud layer retrieval against surface based retrievals.

Quality checks for the ATL09 data are described in Appendix B1. The homogenisation process transforms the atmospheric layer boundaries reported – pairs of cloud top and cloud base heights – into a categorised feature mask distinguishing between

clear sky, cloud, and attenuated regions where the presence of atmospheric scatterers cannot be determined. Vertical profiles of the feature mask are subset according to the co-location criteria (outlined in Sect. 3.3). The feature mask is then horizontally averaged across all the remaining profiles to produce vertical profiles of cloud fraction. The VCF profiles are then homogenised by being vertically interpolated onto a set of $50$ height coordinates with a vertical spacing of $240$ m.

## 3.2 Cloudnet

The Cloudnet retrieval (Illingworth et al., 2007) produces products that categorise the atmospheric profile above a given observatory by optimally combining available retrievals from multiple data sources. Cloudnet synthesises data from ground-based radar, lidar, microwave radiometer, and weather forecast models to produce fields of macrophysical and microphysical quantities such as temperature, cloud occurrence and ice water content. There are $28$ main Cloudnet sites, with numerous campaigns and ARM sites also contributing data. For this study, we use data from four observatories: Ny-Ålesund, Hyytiala, Jülich and

Munich (Ebell et al., 2025). The location of each site is outlined in Table 1.

To produce the homogenised VCF profiles used in our analyses, we start with the Cloudnet *categorize* product, which holds the calibrated synthesised data (accessed through the Cloudnet FMI website, Ebell et al., 2025). Following the definition of cloud mask in the code presented in Tukiainen et al. (2020), we extract the cloud mask as the feature mask used in the analysis. The full quality check process is described in Appendix B2. The vertical profiles of the feature mask are then subset according

to the temporal co-location criteria described in Sect. 3.3. Like the ATL09 homogenisation process, the Cloudnet-derived





feature masks are horizontally averaged to produce profiles of vertical cloud fraction, and vertically interpolated onto a set of 50 height coordinates with a vertical spacing of $240\,\mathrm{m}$.

### 3.3 Co-location scheme

Data from Cloudnet and ATL09 are co-located using the co-location scheme shown in Fig. 1a. The spatial co-location scheme
treats each Cloudnet site as a 0-dimensional point-like source. The ATL09 data then constitutes three distinct 1-dimensional line-like sources – one for each ATLAS strong beam. For each vertical profile in each of the three beams from the ATL09 data, here indexed with subscript $j$, the great-circle distance between the profile's footprint on the ground, and the location of the Cloudnet site, $r_j$, is calculated. The criteria for accepting the ATL09 data is

$$r_j \leq R, \tag{4}$$

such that all profiles falling within a distance $R$ across the Earth's surface of the Cloudnet site are kept in the analysis.

The temporal co-location scheme first requires finding the time of closest approach between ICESat-2 and the Cloudnet site, $t_0$. This is simply the time associated with an ATL09 profile $j$ that minimises $r_j$. That is,

$$t_0 = t_{j'} \,|\, j' = \arg\min_j(r_j). \tag{5}$$

To subset the Cloudnet data, for each Cloudnet profile with index $l$, the criteria applied is

$$|t_l - t_0| \leq \frac{\tau}{2}, \tag{6}$$

where $t_l$ is the time associated with the given Cloudnet profile $l$. This subsets the Cloudnet data based on the profile being recorded within a temporal window of duration $\tau$, centred on the time of closest approach. Thus, the co-location of the ATL09 and Cloudnet data can be parametrised as $\boldsymbol{p} = (R, \tau)$.

Figure 4 shows a demonstration of the co-location of ATL09 and Cloudnet data at Ny-Ålesund. In the ATL09 granule,
ICESat-2 ground tracks passed within $18\,\mathrm{km}$ of the Cloudnet observatory at Ny-Ålesund. Figure 4a shows the locations of the ATLAS strong-beam footprints in relation to the Cloudnet site as ICESat-2 travelled from north to south.

Figure 4b shows the ATL09 cloudmask from the granule. The observed clouds are optically thick enough to attenuate the ATLAS lidar beam throughout the co-location event, so lower level cloud layers will be missed. Layers are detected across a range of heights, with cloud tops varying from $2\,\mathrm{km}$ up to $8\,\mathrm{km}$. Figure 4c plots the spatial co-location criteria from Eq.
(4) as the satellite travels near Ny-Ålesund. The distance from the lidar beam to the ground forms a hyperbolic curve with a minimum separation of $18.4\,\mathrm{km}$. This results in the volume of ATL09 data per overpass being asymptotically linear in $R$. The horizontal dashed line represents the spatial co-location parametrisation of $R = 125\,\mathrm{km}$, and the vertical dashed lines seen in both Figures 4b–c are the boundaries of the subset data based on the spatial co-location criteria, with hatching showing the data rejected by the co-location scheme.

Similarly, Figures 4d–e show the feature mask and temporal co-location criteria applied to the Cloudnet data during the overpass. Early profiles in the mask show low cloud layers between 1-2 km, growing into a thicker cloud layer ranging from





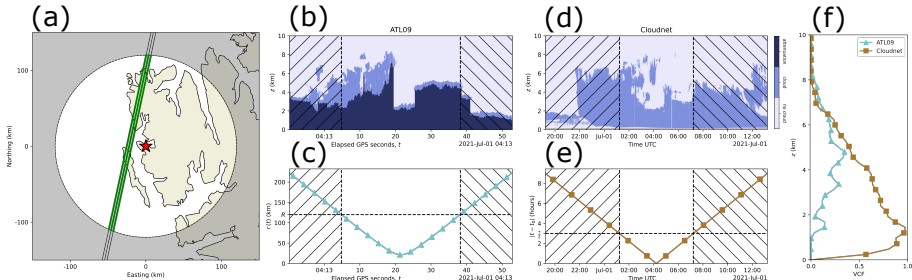

**Figure 4.** The co-location of ATL09 and Cloudnet data at Ny-Ålesund, with ATL09 data from the granule with reference ground track 115 on cycle 12 (dated 01 July 2021). **(a)** The Cloudnet observatory (star), and the three ATLAS strong beams (lines). A circle of radius $R = 125$ km is also drawn around the Cloudnet site. **(b)** The feature mask generated from the ATL09 data associated with strong beam 1, and **(c)** the distance between the ATLAS ground track and Cloudnet observatory, showing the co-location criteria subsetting of the cloudmask. Hatched regions are rejected by the co-location, and the darkest regions show where the ATLAS lidar beam is attenuated. **(d)** and **(e)** are the same as in **(b)** and **(c)**, but for the Cloudnet data contributing to the co-location event. **(f)** Vertical cloud fraction profiles for the ATL09 and Cloudnet cloudmasks, subset by the co-location parametrisation $\boldsymbol{p} = (125 \text{ km}, 6 \text{ hours})$.

near the surface up to $8 \text{ km}$ high, with the top height varying throughout the co-location event. The co-location criteria plotted forms a piecewise linear function, the result of this being that the volume of Cloudnet data used in the analysis is linear in $\tau$.

Figure 4f shows the output of the homogenisation process on the ATL09 and Cloudnet data. The feature masks, subset by the

co-location criteria, are horizontally averaged over all included vertical profiles, producing profiles of vertical cloud fraction. Above $5 \text{ km}$ in height, both VCF profiles visually correlate with each other, indicating that the co-location between the ATL09 and Cloudnet data may be viable. However, below $5 \text{ km}$, the ATL09 VCF values are significantly lower than the Cloudnet VCF values, likely due to attenuation of the ATLAS lidar beam in the higher cloud layers, resulting in lower cloud layers being unobserved by ICESat-2.

## 305 3.4 Mutual information estimation

With the homogenised ATL09 and Cloudnet data both being VCF profiles described on 50 height levels each, the joint probability distribution for pairs of VCF profiles is 100 dimensional. As such, we require a mutual information estimator that accepts multi-dimensional inputs.

The KSG mutual information estimator (Kraskov et al., 2004) and its adaptations to multidimensional data (Holmes and

Nemenman, 2019), $\hat{I}_{\mathrm{KSG}}$, will be used in this work. As inputs, we provide the matched sets of ATL09 and Cloudnet VCF profiles. We use the estimator parameter $k = 10$, as it balances the reduction of estimator variance as $k$ increases with the increased bias resulting from increased $k$. The KSG estimator provides mutual information estimates in units of $\mathrm{nats}$. This is a result of the estimator being derived using natural logarithms instead of logarithms with base 2. The conversion from $\mathrm{nats}$ to the more widely used unit $\mathrm{bits}$ is a scaling factor of $(\ln 2)^{-1} \, \mathrm{bits} \, \mathrm{nat}^{-1}$.



The variance of the KSG estimator, $\sigma_{\mathrm{KSG}}^2$, can be estimated as following the form (Holmes and Nemenman, 2019, Appendix 1)

$$\sigma_{\mathrm{KSG}}^2(N) = \frac{B}{N},\tag{7}$$

where $N$ is the number of samples used in estimating $\hat{\mathrm{I}}_{\mathrm{KSG}}$, and $B$ is a constant parameter to be evaluated. In the maximum likelihood estimation of $B$, we produce $n_i = 10$ non-overlapping partitions of the original data to compute $\sigma_{\mathrm{KSG},i}^2$, and perform this $n_{\mathrm{repeats}} = 20$ times to evaluate $B$.

As well as identifying the optimised parametrisation $\hat{\boldsymbol{p}}$, we also identify regions of the parameter space where $\hat{\mathrm{I}}_{\mathrm{KSG}}(\boldsymbol{p})$ are consistent with the value of $\hat{\mathrm{I}}_{\mathrm{KSG}}(\hat{\boldsymbol{p}})$ (the maximum estimated mutual information), giving a region with finite extent from which an optimised parametrisation could feasibly be selected. To do this, we perform an unequal variances (Welch's) t-test for each $\boldsymbol{p} \neq \hat{\boldsymbol{p}}$, with the null hypothesis being that $\hat{\mathrm{I}}_{\mathrm{KSG}}(\boldsymbol{p}) = \hat{\mathrm{I}}_{\mathrm{KSG}}(\hat{\boldsymbol{p}})$. If the null hypothesis is not rejected with a significance of $0.05$, we consider $\boldsymbol{p}$ to be a possible candidate for an optimised parametrisation.

### 3.5 Validation metrics and methodology

Once we have evaluated the optimised co-location parametrisations for each Cloudnet observatory, we perform a basic comparison of the co-located ATL09 and Cloudnet VCF profiles to demonstrate the impact of using a co-location parametrisation with maximised mutual information instead of other choices of parametrisation.

We compute confusion matrices classifying VCF values into three categories: containing no cloud (nc), when $\mathrm{VCF} = 0$; being partially cloudy (pc), when $0 < \mathrm{VCF} < 1$; and being totally cloudy (tc), when $\mathrm{VCF} = 1$. We make the distinction between nc, pc and tc cases, as VCF values are defined on the closed interval $[0, 1]$, but the probability distribution has degeneracies at 0 and 1, when scenes with no or total cloud cover happen with finite probability. This results in the probability distribution of VCF values having Dirac-delta like contributions at 0 and 1, but being otherwise continuous on the open interval $(0, 1)$.

Having computed confusion matrices, we then compute copula densities between pairs of VCF values across all co-location events and heights within VCF profiles. A copula is the multidimensional extension of the cumulative distribution function for multiple random variables. Random variables are transformed by the probability integral transform – that is, values $x$ for a random variable $X$ with cumulative distribution function $F_X(x)$ are transformed into the variable $U$ with values $u = F_X(x)$, such that $U$ is uniformly distributed on the interval $[0, 1]$. In the bivariate case, with variables $X$ and $Y$, transformed into the variables $U$ and $V$ respectively, the copula is computed as

$$C(u, v) = \mathbb{P}\left(U \leq F_X(x), V \leq F_Y(y)\right),\tag{8}$$

which represents the probability that both uniformly distributed variables $U$ and $V$ are less than their respective coordinates at the same time. Because the marginal distributions of all variables contributing to a copula are uniform, the structure of the copula captures the dependency structure between the variables, independent of the marginal distributions of the original random variables.





In the same way that a probability density function can be obtained by differentiating a cumulative distribution function, so too can a copula density function be obtained by repeated differentiation of the copula. In the bivariate case,

$$c(u,v) = \frac{\partial^2 C(u,v)}{\partial u \, \partial v}.$$ (9)

For independent variables, the copula is given as $C_{\text{independent}}(u,v) = uv$. From this, we derive the independent copula density as $c_{\text{independent}}(u,v) = 1$ uniformly. Thus, we can interpret copula densities greater than 1 as giving pairs $(u,v)$ (and by extension $(x,y)$) that are sampled more frequently than if the variables $U$ and $V$ were independent. Conversely, copula densities less than 1 indicate regions of $(u,v)$ that are sampled less frequently than if the variables were independent. Schölzel and Friederichs (2008) provide a good introduction to methods and interpretation concerning copulae.

Copula densities with values further from 1 indicate that the underlying distribution is dissimilar to the independent joint distribution, which is the desired quality of the co-location. We define the root mean squared difference (RMSD) as a metric of the difference between a copula density and the independent copula:

$$\text{RMSD} = \left( \int_{[0,1]^2} du \, dv \, (c(u,v) - 1)^2 \right)^{\frac{1}{2}}.$$ (10)

Larger RMSD values indicate that a given copula density differs more from the independent copula. If the ATL09 and Cloudnet VCF measurements are entirely independent, c=1 uniformly and the RMSD is zero. If there is dependency between the VCF distributions, then certain pairs of $(u,v)$ values will be sampled more frequently than if the VCF distributions were independent ($c(u,v) > 1$) and, to conserve probability, some pairs $(u,v)$ will be sampled less frequently than if the distributions were independent ($c(u,v) < 1$). The more dependent the VCF distributions are, the larger the area of $(u,v)$ pairs for which $c(u,v) \neq 1$ will be, and the larger the magnitudes of $c(u,v) - 1$ will be in these areas. Thus, stronger dependency between the distributions will result in larger RMSD values. The RMSD attains a maximum value of 1 when $c(u,v)$ describes a one-to-one mapping (e.g. $c(u,v) = \delta(u-v)$).

For this analysis, with a given parametrisation $\boldsymbol{p}$, we identify all co-location events $i$, and compute $\text{VCF}_{\text{ATL09},i}(z)$ and $\text{VCF}_{\text{Cloudnet},i}(z)$. We keep pairs of VCF values for each height $z$, and each co-location event if both VCF values are categorised as pc. We do this so that a valid copula density function can be defined without the degeneracies induced by considering cases with no or total cloud cover.

### 3.5.1 Vertical bias distributions

We compute the vertical distribution of bias between the ATL09 and Cloudnet VCF profiles as a function of height, $\rho_{\text{bias}}(\nu, z)$, where $\nu = \text{VCF}_{\text{ATL09}} - \text{VCF}_{\text{Cloudnet}}$ is the bias, bounded between $-1$ and $1$. The expected bias and variance of the bias as a



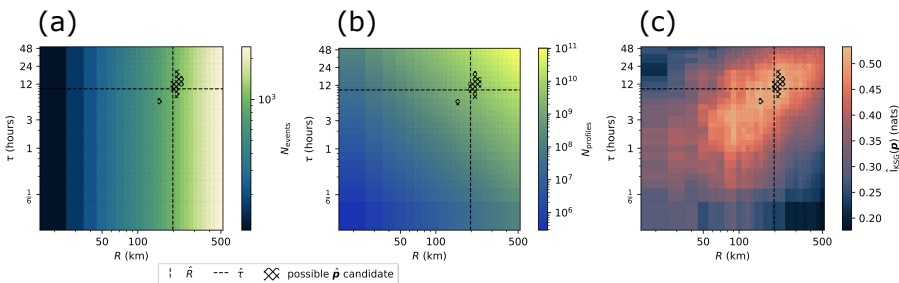

**Figure 5.** The number of co-location events **(a)**, pairwise vertical profile comparisons **(b)**, and the mutual information **(c)** computed between ATL09 data and Cloudnet data from the observatory at Jülich, as a function of co-locations parametrisation $\boldsymbol{p} = (R, \tau)$. The maximum mutual information (indicated by crossing dashed lines) occurs at $\hat{\boldsymbol{p}} = (196.9\,\mathrm{km}, 10\,\mathrm{hours})$. Hatching denotes regions of parameter space where $\hat{\mathrm{I}}_{\mathrm{KSG}}(\boldsymbol{p})$ is not significantly different from $\hat{\mathrm{I}}_{\mathrm{KSG}}(\hat{\boldsymbol{p}})$.

function of height are calculated as

$$\mathbb{E}[\nu|z] = \int_{-1}^{1} d\nu \, \rho_{\mathrm{bias}}(\nu, z)\nu, \tag{11}$$

$$\mathrm{Var}[\nu|z] = \int_{-1}^{1} d\nu \, \rho_{\mathrm{bias}}(\nu, z)\nu^2 - \mathbb{E}[\nu|z]^2. \tag{12}$$

### 3.6 Results

#### 3.6.1 Case study: Jülich

We will start by demonstrating the results of the mutual information computation at a singular site, Jülich, before showing the results across all four example sites.

Figure 5a shows the number of co-location events used in the study as a function of the co-location parametrisation. The Cloudnet data at Jülich forms a near-complete record, meaning co-location events are solely dependent on the availability of sufficiently close ICESat-2 orbital tracks. Thus, we expect no gradient in $N_{\mathrm{events}}$ as a function of $\tau$, which we see.

We should expect the number of co-location events to be approximately a linear function of $R$. At a given latitude, orbital ground tracks can be split into two sets, ascending and descending. Within each set, all orbital tracks are approximately parallel at a given latitude, so the number of events included in the analysis is the number of orbital tracks intersecting a line centred at Jülich, of length $2R$, perpendicular to the orbital tracks. Given the rotational symmetry of repeating orbital tracks, the across-track density of orbits, $\rho_{\mathrm{orbits}}$, can be approximated as constant at given latitude, and an equation approximating $\rho_{\mathrm{orbits}}$ is given in Appendix C. The result is $N_{\mathrm{events}}$ being approximately proportional to $\rho_{orbits}R$. Given the logarithmic scaling of the colour map, and the plot coordinates, the smooth colour gradient seen in Fig. 5a indicates that $N_{\mathrm{events}}$ is a polynomial function of $R$, consistent with the above arguments.





Figure 5b shows the number of pairwise vertical profile comparisons, $N_{\text{profiles}}$, computed as

$$N_{\text{profiles}} = \sum_i n_{\text{ATL09},i} \; n_{\text{Cloudnet},i}, \tag{13}$$

where $i$ represents a co-location event, and $n_{s,i}$ is the number of vertical profiles from data source $s$, in co-location event $i$, included in the analysis after the application of the co-location criteria. This can be thought of as the volume of data being

compared – if the homogenisation process was not to aggregate the data but instead compare individual observations in a pairwise fashion, this is the number of paired VCF profiles that would be available in the analysis.

As with $N_{\text{events}}$, the smooth colour gradient in Fig. 5b indicates $N_{\text{profiles}}$ is approximately polynomial as a function of $R$ and $\tau$. We expect $N_{\text{profiles}}$ to be proportional to $\tau$, as for each co-location event, the number of included vertical profiles linearly scales with the duration of the time window of length $\tau$. We expect $N_{\text{profiles}}$ to be proportional to $R^2$, one power coming from

a proportionality to $N_{\text{events}}$, and the other deriving from the fact that for each given co-location event, the number of included vertical profiles scales as a function of $\cosh(R)$, which is linearly proportional to $R$ in the limit of large $R$ values. The results are consistent with $N_{\text{profiles}} \propto R^2 \tau$.

Figure 5c shows the mutual information calculated according to Sect. 3.4 across all tested parametrisations. Mutual information values range from a minimum of $0.177 \pm 0.010$ nats at $\boldsymbol{p} = (500\,\text{km}, 300\,\text{s})$, to a maximum of $0.533 \pm 0.020$ nats

with $\hat{\boldsymbol{p}} = (196.9\,\text{km}, 10\,\text{hours})$. The mutual information surface is unimodal, with a ridge of higher values where the global maximum is found. As $\boldsymbol{p}$ moves away from $\hat{\boldsymbol{p}}$, the mutual information appears to decrease monotonically. In the region of lower $R$ and $\tau$ values, this can be explained as the mutual information estimator being data limited. The KSG estimator acts on the pairs of VCF profiles as though they are drawn from a 100-dimensional joint probability distribution. In order to learn structure in this joint distribution and compute larger mutual information values, a sufficient number of co-location events must

contribute to the analysis, so that the 100-dimensional distribution can be sampled densely enough to infer the structure.

For larger values of $R$ and $\tau$, we expect the rate of errors as a results of co-location mismatch to increase. This will contaminate the VCF comparisons with uncorrelated and independent profile comparisons. As is shown in Appendix A, the inclusion of independent data to the comparison necessarily decreases the possible upper bound of the mutual information. The $\hat{\mathrm{I}}_{\text{KSG}}(\boldsymbol{p})$ surface seen in Fig. 5c is consistent with these expectations.

There is a region of $\boldsymbol{p}$ near $\hat{\boldsymbol{p}}$ for which $\hat{\mathrm{I}}_{\text{KSG}}(\boldsymbol{p})$ are not significantly different from $\hat{\mathrm{I}}_{\text{KSG}}(\hat{\boldsymbol{p}})$ with a significance of $0.05$, indicated by the hatching on Fig. 5c. This region represents other possible choices of an optimal parametrisation that provides similarly informative comparisons between the ATL09 and Cloudnet VCF retrievals at Jülich. The region predominantly exists for values of $R \geq \hat{R}$, extending as far as $\sim 240\,\text{km}$. The possible values of $\tau$ range from $\sim 8$ hours (less than $\hat{\tau}$) to $\sim 18$ hours (greater than $\hat{\tau}$). As seen in Fig. 5a-b, the hatched regions represent similar or larger input data volumes when compared to $\hat{\boldsymbol{p}}$,

indicating that the KSG estimator could be not data-limited prior to incorporating sufficient independent data to contaminate the results, such that $\hat{\mathrm{I}}_{\text{KSG}}$ has attained its upper bound given the data distributions. There is also a smaller region of possibly optimised parametrisations found near $\boldsymbol{p} = (150\,\text{km}, 6\,\text{hours})$, with lower associated data volumes.





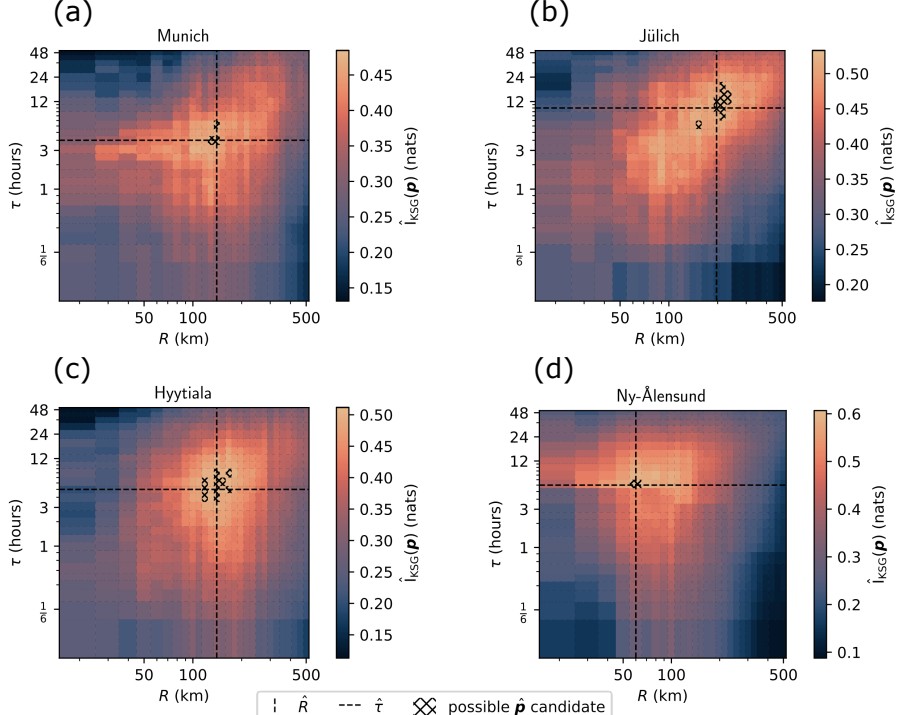

**Figure 6.** The mutual information computed between ATL09 and Cloudnet VCF profiles as a function of co-location parametrisation $\boldsymbol{p} = (R, \tau)$ for the Cloudnet observatories at Munich **(a)**, Jülich **(b)**, Hyytiala **(c)** and Ny-Ålesund **(d)**. The location of the maximum mutual information value, $\hat{\boldsymbol{p}}$, is indicated by the crossing dashed lines, and hatching indicates regions in parameter space where $\hat{I}_{\mathrm{KSG}}(\boldsymbol{p})$ does not differ significantly from $\hat{I}_{\mathrm{KSG}}(\hat{\boldsymbol{p}})$. Panel **(b)** is the same as Fig. 5c.

### 3.6.2 Mutual information at four Cloudnet observatories

We will start by considering the $\hat{I}_{\mathrm{KSG}}(\boldsymbol{p})$ surfaces for Munich, Jülich and Hyytiala, as seen in Fig. 6a–c. Qualitatively, all

three sites show a similar structure of a unimodal surface, a ridge of higher mutual information values that contains a single global maximum. These similarities in structure can be explained using the same arguments as in Sect. 3.6.1, with regions in the parameter space where the KSG estimator is data limited, and regions where the input data to the estimator is contaminated with independent samples. Despite structural similarities, the optimised parametrisations, $\hat{\boldsymbol{p}}$, at each site differ, as do the magnitudes of $\hat{I}_{\mathrm{KSG}}(\hat{\boldsymbol{p}})$. Values for $\hat{\boldsymbol{p}}$, $\hat{I}_{\mathrm{KSG}}(\hat{\boldsymbol{p}})$, and other quantities at each Cloudnet observatory are given in Table 1.

Table 1 shows that the optimised radius $\hat{R}$ can vary on a per-site basis. For example, $\hat{R}_{\mathrm{Jülich}}$ is larger than both $\hat{R}_{\mathrm{Hyytiala}}$ and $\hat{R}_{\mathrm{Munich}}$. One possible explanation for this is the relatively flat orography around the Jülich Cloudnet observatory, when compared (for example) to the proximity of the Munich observatory to the Alps. The mountainous orography of the Alps could result in smaller spatial scales over which local cloud formation is correlated, giving rise to smaller spatial informativity scales $\hat{R}$ than at other locations like Jülich. The values of $\hat{\tau}$ at Munich, Jülich and Hyytiala are similar orders of magnitude, ranging





**Table 1.** The locations of the Cloudnet sites used in the analysis, and important results of the mutual information calculation between the ATL09 and Cloudnet VCF profiles at each site. $\rho_{\text{orbits}}$ represents the normalised across-track density of ICESat-2 orbits at the latitude of the Cloudnet site. $\hat{\boldsymbol{p}} = (\hat{R}, \hat{\tau})$ represents the optimised parametrisation at which the maximum mutual information, $\hat{\text{I}}_{\text{KSG}}(\hat{\boldsymbol{p}})$, is found. $N_{\text{events}}(\boldsymbol{p})$ is the number of co-location events from which data is included with a parametrisation $\boldsymbol{p}$. $N_{\text{profiles}}(\boldsymbol{p})$ is the number of pairwise profile comparisons made between ATL09 and Cloudnet VCF profiles across all co-location events for a given parametrisation $\boldsymbol{p}$.

| site | latitude (°N) | longitude (°E) | $\rho_{\text{orbits}}$ | $\hat{R}$ (km) | $\hat{\tau}$ (hours) | $N_{\text{events}}(\hat{\boldsymbol{p}})$ | $N_{\text{profiles}}(\hat{\boldsymbol{p}})$ | $\hat{\text{I}}_{\text{KSG}}(\hat{\boldsymbol{p}})$ (nats) |
|---|---|---|---|---|---|---|---|---|
| Ny-Ålensund | 78.9 | 11.9 | 5.29 | 60.0 | 6.0 | 932 | 6.06e+08 | 0.607 ±0.020 |
| Hyytiala | 61.8 | 24.3 | 2.12 | 140.3 | 5.0 | 881 | 1.01e+09 | 0.511 ±0.016 |
| Jülich | 50.9 | 6.4 | 1.59 | 196.9 | 10.0 | 956 | 3.27e+09 | 0.533 ±0.020 |
| Munich | 48.1 | 11.6 | 1.50 | 140.3 | 4.0 | 634 | 6.60e+08 | 0.484 ±0.018 |

from 4 hours to 10 hours. These values are consistent with the temporal scale of cloud evolution found in other studies (Shupe, 2011; Silber et al., 2018).

At Ny-Ålesund, the $\hat{\text{I}}_{\text{KSG}}(\boldsymbol{p})$ surface (see Fig. 6d) shares some qualities with the mutual information surfaces seen at the other Cloudnet observatories – the surface is unimodal, with $\hat{\text{I}}_{\text{KSG}}(\boldsymbol{p})$ decreasing as $\boldsymbol{p}$ moves away from $\hat{\boldsymbol{p}}$, and has a value of $\hat{\tau}_{\text{Ny-Ålesund}} = 6$ hours, which is in the same order of magnitude as the values of $\hat{\tau}$ at the other Cloudnet sites. The mutual

information values increase sharply as $R$ increases above $R \sim 50$ km and as $\tau$ increases above $\tau \sim 5$ hours. This sharp increase in the mutual information values results in the maximum $\hat{\text{I}}_{\text{KSG}}(\hat{\boldsymbol{p}}) = 0.607 \pm 0.020$ nats occurring at $\hat{\boldsymbol{p}} = (60\,\text{km}, 6\,\text{hours})$. The value of $\hat{R}_{\text{Ny-Ålesund}}$ is lower than those found at the other three Cloudnet sites. This could partially be explained by the orography around Ny-Ålesund. The island of Spitsbergen, on which Ny-Ålesund is sat, is mountainous with peaks as high as 1700 m. The proximity of the Cloudnet observatory at Ny-Ålesund to the mountainous orography could lead to a reduced

spatial autocorrelation scale between the clouds observed at the Cloudnet site and those to the East, that may be observed by ICESat-2. Thus, physically uncorrelated VCF comparisons would contaminate the mutual information calculation at smaller values of $R$ than at other sites, reducing the value of $\hat{R}$.

Another more subtle effect impacts the value of $\hat{R}$ at each site. For a given parametrisation $\boldsymbol{p}$, the KSG mutual information estimator accepts $N_{\text{events}}(\boldsymbol{p})$ pairs of VCF profiles in order to estimate $\hat{\text{I}}_{\text{KSG}}(\boldsymbol{p})$. $N_{\text{events}}(\hat{\boldsymbol{p}})$ is of a similar order of magnitude

at Ny-Ålesund when compared to Hyytiala and Jülich, even with a substantially smaller value of $\hat{R}$. This is due to the local across-track density of orbits being an increasing function of latitude. As well as being linearly proportional to $R$, $N_{\text{events}}$ has a a functional dependency on latitude, which is derived in Appendix C. The normalised across-track orbital density, $\rho_{\text{orbits}}$, is given for each site in Table 1. The higher local density of orbits at Ny-Ålesund compared to the other sites allows for more data to be used in the estimation of $\hat{\text{I}}_{\text{KSG}}(\boldsymbol{p})$ at Ny-Ålesund than at Cloudnet observatories at lower latitudes. This could result

in denser sampling of the 100-dimensional joint probability distribution at lower values of $R$ for more poleward locations. Thus, the mutual information estimator, being able to infer structure in the joint probability distribution at smaller values of $R$,





switches from a data limited regime to being sensitive to the inclusion of independent VCF samples. This could result in the estimated value $\hat{\mathrm{I}}_{\mathrm{KSG}}(\boldsymbol{p})$ being reduced from the maximum attainable value at lower values of $R$ at Ny-Ålesund than at other sites, as the structure has already been inferred by the KSG estimator, and the effect of contamination by independent data

outweighs the inclusion of more VCF comparisons that may be related.

The hatched regions in Fig. 6 are unique across all four Cloudnet observatories, but can be split into two sets: Jülich and Hyytiala, being generally surrounded by flatter orography and having larger plausible extents in parameter space from which optimised co-location parametrisations can be selected, and; Munich and Ny-Ålesund, having much closer proximity to mountainous orography, and having substantially smaller regions of parameter space from which a plausibly optimised parametrisa-

tion can be selected – in the case of Ny-Ålesund, only a single tested parametrisation, specifically $\hat{\boldsymbol{p}}_{\mathrm{Ny\text{-}Ålesund}}$, can significantly be considered optimised. This split between the two sets suggests not only are the optimised parametrisations $\hat{\boldsymbol{p}}$ different between locations, but that the co-locations at each site are uniquely sensitive to the choice of $\boldsymbol{p}$, with the sites located closest to mountainous orography being the most sensitive to the choice of co-location parametrisation.

By quantifying the mutual information encoded between our data, we learn where and when we should be selecting data

around each Cloudnet observatory, and find that the spatial and temporal scales for data subsetting are different at each location. In identifying $\hat{\boldsymbol{p}}$, we are able to analyse the maximum volume of data while minimising the contamination of the results through the inclusion of independent data. We have demonstrated that the value of $\hat{\boldsymbol{p}}$ is influenced by local factors, such as mountainous orography near the surface-based observatories, and non-local factors such as the satellite sampling strategy. The non-trivial shape of the $\hat{\mathrm{I}}_{\mathrm{KSG}}(\boldsymbol{p})$ surfaces computed for each Cloudnet observatory show that optimising the parametrisation requires a

full exploration of the parametrisation space, and that optimising each individual parameter independently will not adequately identify the true maximum in the estimated mutual information. Moreover, we have shown that a one-size-fits-all approach to selecting the co-location parametrisation is unsuitable. Using $\hat{\boldsymbol{p}}_{\mathrm{Ny\text{-}Ålesund}}$ at the other Cloudnet observatories would reduce the number of permitted co-location events, reducing the data volume available for the subsequent analyses. Instead, if we chose $\hat{\boldsymbol{p}}_{\mathrm{Munich}}$ for all Cloudnet observatories, the co-location at Ny-Ålesund would be degraded by the inclusion of independent

samples, but the co-location at Jülich would conversely be degraded by a reduction in the available co-location events.

### 3.6.3    Dependency between ATL09 and Cloudnet VCFs for different co-location parametrisations

To demonstrate the importance of the choice of co-location parametrisation $\boldsymbol{p}$ on the validation of satellite data, we compute confusion matrices and copulae between all pairs of VCF values in the ATL09-Cloudnet VCF profile pairs for the parametrisations given in Table 2. Two canonical choices of co-location parametrisation exist. Firstly, one could choose to only accept

co-locations that minimise the co-location mismatch between the data sources, at the expense of reducing the available data volume for subsequent analysis. This is represented by $\boldsymbol{p}_{00} = (50\,\mathrm{km}, 30\,\mathrm{minutes})$. The second canonical choice is to use all of the available data, in the hopes of having enough good data comparisons that the inclusion of independent data is not impactful. This is represented by $\boldsymbol{p}_{11} = (500\,\mathrm{km}, 2\,\mathrm{days})$.

Many previous bodies of work use the same co-location scheme as outlined in Sect. 3.3. Many of these studies use values

of $R$ that are low integer multiples of $50$ km, and similarly values of $\tau$ that are low integer multiples of $30$ minutes (e.g.



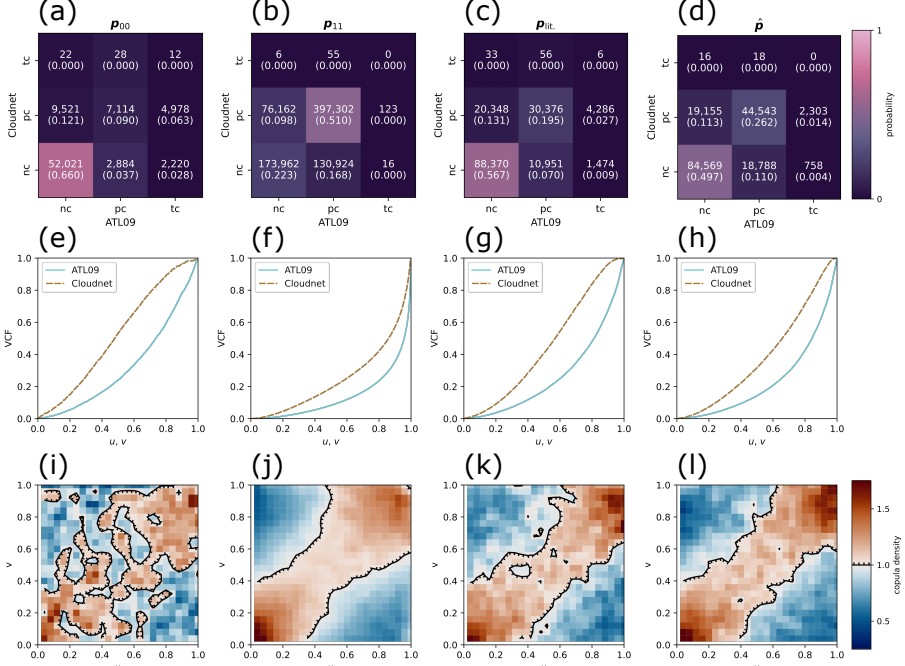

**Figure 7.** Confusion matrices for the detection of no cloud (nc), partial cloud (pc) and total cloud (tc) across all VCF values at all sites for the co-locations representing: a data limited co-location, $p_{00}$ **(a)**; an independent data contaminated co-location, $p_{11}$ **(b)**; a co-location typical of those in the literature, $p_{\text{lit.}}$ **(c)** and; the per-site optimised co-location, $\hat{p}$ **(d)**. **(e–h)** Cumulative distribution functions for ATL09 and Cloudnet VCF values across all sites for associated co-location parametrisations, conditional that the VCF values are strictly between 0 and 1 to remove degeneracies in the copula densities. **(i–l)** Copula density plots for the associated co-location parametrisations. The contour indicates copula densities of 1, with higher densities indicating regions in $(u, v)$ space that are sampled more frequently than if the variables $U$ and $V$ are independent.

Robinson et al., 2025; Lin et al., 2022; Protat et al., 2009, 2014a; Schuster et al., 2012; Baars et al., 2023; Wang and Stammes, 2014; Pappalardo et al., 2010; Mona et al., 2009; Proestakis et al., 2019; Papagiannopoulos et al., 2016; Liu et al., 2017; Lu et al., 2021; Pauly et al., 2019; Mamouri et al., 2009). As such, we use $p_{\text{lit.}} = (100\,\text{km}, 3\,\text{hours})$ to represent a typical choice of co-location parametrisation from the literature.

The parametrisation $\hat{p}$ represents the collection of all the co-location events across the four Cloudnet observatories, using each site-specific optimised co-location parametrisation (see Table 1).

     Confusion matrices for the retrieval of no cloud (nc), partial cloud (pc) and total cloud (tc) VCF values between the ATL09 and Cloudnet data are given in Fig. 7a–d. In all tested parametrisations, the cells corresponding to (nc, nc) and (pc, pc) are the two most probable states. The accuracy, being the sum of the confusion matrix diagonal elements where both retrievals agree, is given as ACC in Table 2. Across the tested parametrisations, the accuracy ranges between 0.73 to 0.76, with $p_{\text{lit.}}$ having the

highest accuracy of 0.762. The contaminated-data regime given by $p_{11}$ has the lowest proportion of data falling in the (tc, tc)





**Table 2.** Results from the computation of copulae comparing VCF values between ATL09 and Cloudnet data for all the tested parametrisations. The accuracy of the agreement between the VCF retrievals for the categories nc, pc and tc (see Fig. 7) is given as ACC. $c_{\min}$ is the minimum copula density for the given parametrisation, $c_{\max}$ is the maximum achieved copula density, and $c(1,1)$ is the discretised tail dependence of the copula. RMSD is the root mean squared difference of the copula density from the independence copula density. Values in bold indicate the best parametrisation for the given metric (the notion of best being defined in the text).

| parametrisation | $R$ (km) | $\tau$ (hours) | ACC | $c_{\min}$ | $c_{\max}$ | $c(1,1)$ | RMSD |
|---|---|---|---|---|---|---|---|
| $\boldsymbol{p}_{00}$ | 50.0 | 0.5 | 0.751 | 0.44 | 1.65 | 1.05 | 0.21 |
| $\boldsymbol{p}_{11}$ | 500.0 | 48 | 0.734 | 0.49 | 2.09 | 0.91 | 0.22 |
| $\boldsymbol{p}_{\text{lit.}}$ | 100.0 | 3 | **0.762** | 0.50 | 1.86 | 1.21 | 0.24 |
| $\hat{\boldsymbol{p}}$ | (see Table 1) | | 0.759 | **0.41** | **2.27** | **1.70** | **0.27** |

classification, and the highest proportion of data falling into the (pc, pc) classification. This is a result of the large integration scales for the larger $R$ and $\tau$ values, decreasing the probability that all or none of the vertical profiles contain any cloud at a given height. $\boldsymbol{p}_{\text{lit.}}$ and $\hat{\boldsymbol{p}}$ produce similar confusion matrices, although $\hat{\boldsymbol{p}}$, typically having larger $R$ and $\tau$ values than $\boldsymbol{p}_{\text{lit.}}$, has
less degeneracy in the VCF values and as a result has a higher proportion of (pc, pc) samples than $\boldsymbol{p}_{\text{lit.}}$.

Figures 7e–h show the cumulative distribution functions transforming VCF values to their uniformly distributed copula coordinates. These are defined only for the data falling in the (pc, pc) classification so as to avoid degeneracies in the functions. The shapes of the curves indicate that in all cases, the density of ATL09 VCF samples decreases as a function of the ATL09 VCF value, concentrating the majority of samples at lower values. For $\boldsymbol{p}_{00}$ and $\boldsymbol{p}_{\text{lit.}}$, the Cloudnet distribution functions are
slightly inflected, as a result of having more VCF samples close in value to 1 than the parametrisations $\boldsymbol{p}_{11}$ and $\hat{\boldsymbol{p}}$. This is likely due to the smaller $\tau$ values for $\boldsymbol{p}_{00}$ and $\boldsymbol{p}_{\text{lit.}}$ making it more probable that VCF values closer to 1 are possible.

Figures 7i–l show the copula densities for the tested parametrisations. $\boldsymbol{p}_{00}$, being data-limited, has a noisy copula density surface. The other parametrisations all have comparably smooth surfaces, with a well defined ridge of higher densities around the line $u = v$. This shows that for $\boldsymbol{p}_{11}$, $\boldsymbol{p}_{\text{lit.}}$ and $\hat{\boldsymbol{p}}$, when ATL09 retrieves a given VCF value, the Cloudnet VCF value typically
trends with that value – albeit in a non-linear fashion due to the non-equal cumulative distribution functions.

The copula density associated with $\boldsymbol{p}_{11}$ has the smoothest appearance, due to having the highest number of contributing samples it is generated from. Despite this, the upper-right corner shows that $c(1,1) = 0.91$, with $\boldsymbol{p}_{11}$ being the only parametrisation where the generated copula shows that the ATL09 and Cloudnet joint retrieval of the most extreme cloud fractions is sampled less frequently than if the retrievals were independent. This is an undesirable characteristic in the comparison of the retrievals,
and hints at the contamination of the comparison by VCF profiles that are independent due to the large spatio-temporal domain within which the co-locations happen. Considering the value of $c(1,1)$ for different $\boldsymbol{p}$, $\hat{\boldsymbol{p}}$ produces the copula with the highest density in the upper-right tail.





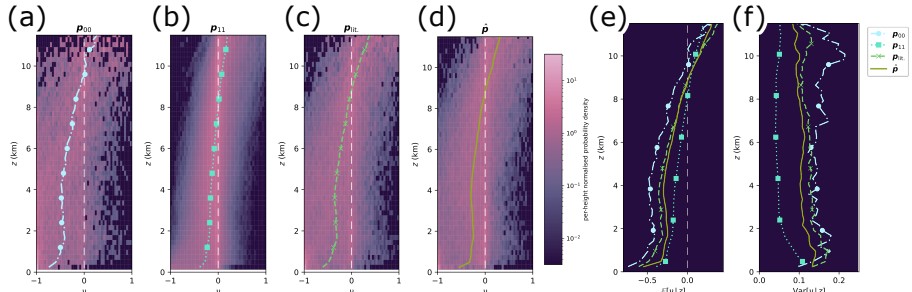

**Figure 8.** Bias distributions between ATL09 and Cloudnet VCF profiles as a function of height for the parametrisations $p_{00}$ **(a)**, $p_{11}$ **(b)**, $p_{\text{lit.}}$ **(c)** and $\hat{p}$ **(d)**. Expected bias profiles are given as coloured dashed lines. **(e)** The expected bias profiles for the different parametrisations plotted together, using the same line styles and markers as in their individual panels. **(f)** The variance in the bias distributions as a function of height.

We can also show that the copula density associated with $\hat{p}$ yields the smallest minimum copula density value, $c_{\min} = 0.41$, and the highest maximum copula density value, $c_{\max} = 2.27$, across the tested parametrisations. The RMSD values indicate

that the $p_{\text{lit.}}$ co-location parametrisation is better than utilising all available data, or limiting the co-locations in order to naively reduce the rate of co-location mismatch. However, we see that $\hat{p}$ yields the copula with the highest RMSD value of the tested parametrisations, with RMSD $= 0.27$.

These results show that using the optimised co-location parametrisation $\hat{p}$ at each Cloudnet site yields a better relationship between the VCF distributions than using other parametrisations, either by including fewer independent samples, or by

including a larger number of dependent samples.

### 3.7 Vertical bias profiles

Figure 8a–d shows the bias distributions between the ATL09 and Cloudnet VCF profiles as a function of height. One common feature across all parametrisations is that the expected bias is negative for heights $< 8$ km, and is positive for higher altitudes ($z > 10$ km). This indicates that the ATLAS lidar is observing more cloud presence than Cloudnet at higher altitudes (and

visa-versa at lower altitudes). This could be explained by the viewing geometry (i.e. ICESat-2 viewing clouds from above and Cloudnet viewing clouds from below) and is consistent with comparisons of other vertically resolved satellite retrievals of cloud presence against surface observations (e.g. McErlich et al., 2021).

Figure 8e shows the expected bias profiles for the shown parametrisations as a function of height. In all cases, the bias is negative for $z < 8$ km indicating that ICESat-2 is less sensitive to clouds at these altitudes than Cloudnet is. Similarly in all

cases, for $z > 10$ km, the biases are positive showing that ATL09 is reporting more cloud at higher altitudes than Cloudnet is. This could be explained by the differences in viewing geometry between the platforms, and the general trend is consistent with results from other comparisons of vertically resolved satellite retrievals of cloud presence against surface based observations (McErlich et al., 2021). Although the expected profiles are all qualitatively similar, the height at which the bias changes sign is





different between the parametrisations. $p_{11}$ has the lowest change from negative to positive bias at a height of $7.9$ km. Above
this, both $p_{\text{lit.}}$ and $\hat{p}$ have bias transition heights around $8.5$ km (their transitions occur within one histogram height bin of each
other). $p_{00}$ has the highest bias transition height of $9.4$ km.

Figure 8f shows the variance of the bias distributions for the different parametrisations as a function of height. $\hat{p}$ has a
consistently lower variance across all heights when compared to $p_{00}$ and $p_{\text{lit.}}$. $p_{11}$ around $z = 0$ has a similar variance to the
other parametrisations, but above this height, the variance reduces to a nearly constant value around $0.2$ for heights above $2$
km.

Simply from observing the expected bias profiles and the variance profiles, one may deduce that the selection of $p_{11}$ gives the
best comparison between the data, as the magnitude of the expected bias and variance are the lowest across all parametrisations.
However, in using the expected bias and the variance to determine the choice of $p$, we have necessarily biased our results to
be closer to *ideal* values. As was shown in Sect. 3.6.3, choosing $p_{11}$ over other parametrisations includes more comparisons of
independent data in the analysis when compared to the choice of $\hat{p}$. If we were to use $p_{11}$, corrective factors for the bias could
be too small in magnitude, and the uncertainty budget of the VCF profiles might be underestimated. Thus, we conclude that
the metrics computed between the data to be compared should not be used to assess the quality of the co-location, and that the
parametrisation should instead be evaluated by maximising the mutual information between the data to be compared.

By incorrectly choosing $p$, we subscribe to two possible outcomes: a degradation in the quality of the results of our com-
parisons between the data, obtaining quantitatively different results due to the difference in the input data, or; quantitatively
similar results to those found when using $\hat{p}$, that may arise as a result of competing erroneous effects due to the inclusion
of independent data, or the rejection of dependent data. With no independent analysis on the choice of $p$, there is no way to
distinguish between good and bad co-locations.

## 4  Discussion

This paper presents a unified framework for determining an optimised parametrisation that should be used when spatiotempo-
rally co-locating geospatial data, before comparative analyses or data synthesis can be performed. We utilise mutual information
as a domain- and data-agnostic metric quantifying the quality of a data co-location, independent of the metrics typically used
in subsequent analyses. Selecting the co-location parametrisation by optimising the comparison metrics of the analysis risks
biasing the validation results to the highest attainable values given the data. As such, the comparison metric cannot be used
to assess the quality of the co-location of the data used to compute the comparison metric. By definition, we do not know the
value of the comparison metric, and by parametrising data co-location to maximise or optimise the comparison metric, we
can be thought of as applying a prior distribution to the validation metric of what value we would like the result to have. Our
framework allows you to assess your data independently from the co-location, reducing the effects of sample bias induced by
a bad co-location.

We have demonstrated for a novel data comparison how the framework can be utilised, defining a co-location scheme to
apply to ICESat-2 ATL09 derived VCFs and Cloudnet derived VCFs. Using a grid search, we were able to identify optimised





co-location parametrisations $\hat{p}$ per Cloudnet observatory, and with a basic comparison between the VCF values, show that this parametrisation produces better comparisons of the data than a typically used parametrisation, as well as naive choices that maximise or minimise the used data volume. Still, there are some important parts of this framework that need addressing.

## 4.1 The choice of mutual information estimator

In this study, we utilised the adaptation of the KSG estimator (Kraskov et al., 2004) proposed by Holmes and Nemenman (2019). We chose this implementation of a mutual information estimator as it allows for the mutual information to be computed between distributions of arbitrary dimension, and the development of variance estimation for the mutual information estimator allows us to determine regions in parameter space within which $\hat{p}$ may lie, as opposed to identifying a single value for $\hat{p}$.

By accepting data or arbitrary dimension, the KSG estimator is widely applicable within the Earth sciences community. Care is needed to properly implement and interpret the outputs of the estimator, but this is the case for all other mutual information estimators.

We believe the KSG estimator is at present a suitable choice of estimator for many problems, but other estimators may be developed, or be shown to be more robust for data meeting certain requirements. In these cases, it is up to the researcher's judgement to decide which estimator is most appropriate for their analysis.

## 4.2 Physical interpretation of $\hat{p}$

Ascribing physical meaning to the values of the parameters in $\hat{p}$ may be tempting, as they define a spatio-temporal region where data falling within the region maximises the mutual information between the Cloudnet and ATL09 VCF retrievals. The values of $\hat{R}$ and $\hat{\tau}$ will be intimately linked to the spatial and temporal scales of cloud evolution at each given Cloudnet site. However, due to the high degree of non-linearity in the mutual information estimation, relating $\hat{R}$ and $\hat{\tau}$ to well-defined concepts such as autocorrelation scales is theoretically challenging. This work does not concern itself with elucidating the physical meaning of the values associated with $\hat{p}$, but further work could allow empirical relationships between the components of $\hat{p}$ and other well-defined quantities to be identified, opening up new methods for the evaluation of thesea different quantities.

As was shown in Sect. 3.6.2, the extent of the regions from which optimised co-location parametrisations can be selected is site dependent, and depends on local factors such as orography, as well as factors relating to the sampling strategy at each site. As well as inferring physical meaning from the parameters in $\hat{p}$, work could be done to model how the plausible region of optimised co-location parametrisations depends on the local environment, which would allow for planning of sampling strategies in advance of satellite missions to capitalise on maximising mutual information at different locations where reference data is recorded.

## 4.3 Choice of co-location scheme

We demonstrated the use of a simple co-location scheme, only considering (spatially) the separation between the ATL09 data and the Cloudnet observatory. Even with this simplified treatment of the spatial distribution of clouds, we were able to show



an improvement of the comparison metrics calculated between the ATL09 and Cloudnet data, simply by choosing to use the co-location parametrisation $\hat{p}$ over other co-location parametrisations.

However, being simplified, the co-location scheme described in Sect. 3.3 still allows comparisons between independent VCF profiles. The scheme could be augmented with additional co-location criteria and parameters, in order to encode more a priori knowledge that constrains the data comparisons being permitted. As an example, Lu et al. (2021) compare CALIPSO cloud layer boundaries against those identified by a ceilometer at the Eastern North Atlantic (ENA) ARM observatory, located on the Azores. As well as subsetting data according to the co-location scheme described in Sect. 3.3, using $p = (150\ \mathrm{km}, 1$

615 hour), co-location events are further subset based on the prevailing wind direction at the time of closest approach, in order to reduce the contamination of the analysis by comparing orographically disturbed cloud layers. The approach introduces two angular windows, one used if CALIPSO passes to the east of the ENA observatory, and one used if CALIPSO passes to the west. Each angular window is defined by two extreme angles, within which if the wind is blowing from within the angles subtended by the window, the co-location event is excluded from the analysis. In Lu et al. (2021), the angles are

620 chosen from cardinal directions, and the windows as a result subtend $90°$ each. In this framework, the angles defining the edges of these windows could each become a free parameter, resulting in the 6-dimensional parameter space given by $p = (R, \tau, \theta_{\mathrm{east,min}}, \theta_{\mathrm{east,max}}, \theta_{\mathrm{west,min}}, \theta_{\mathrm{west,max}})$. The more complicated parametrisation space and co-location criteria may allow for higher mutual information between the datasets to be achieved if there was a systematic shortcoming with the simpler co-location scheme that allows independent samples to be permitted in the analysis regardless of the choice of parametrisation.

In our demonstration, the 2-dimensional parameter space was explored by a grid search method to compute mutual information values across a range of parametrisations. Higher dimensional parameter spaces come at the cost of increasing computational overhead, and grid search methods scale exponentially with the number of dimensions. Thus, identifying optimised parametrisations in higher dimensional parameter spaces may require the use of methods such as stochastic gradient descent (in this case, minimising negative mutual information) to efficiently explore the possible parametrisations and identify

suitable choices of $\hat{p}$.

## 4.4 Applicability of the framework

This framework is widely applicable to problems where the use of multiple data products on non-homogenised coordinates is required. In this study, the framework is demonstrated on the use of a comparison of cloud presence data products. Mutual information can (with the Holmes estimator) be computed between data of arbitrary dimension. Thus, validations of scalar

quantity retrievals (e.g. aerosol optical depth, etc.) and vector quantity retrievals (e.g. VCFs, etc.) are both possible.

Vector quantity retrievals are equivalent to the joint retrieval of two geophysical fields between distinct data sources. For example, the joint retrieval of temperature and relative humidity could be considered a vector quantity and can thus be compared between data sources.

By adapting the co-location scheme, data can be matched between (for example) two different satellite platforms (see Fig.

1c). This is important for facilitating analyses that characterise the differences between the retrievals of the same quantities by different satellites, better characterising uncertainties induced in long-term records of geophysical quantities.





The focus of the study need not be comparisons of satellite retrievals. The data could compare data from other mobile platforms, such as planes and ships. Outputs from generalised circulation models could be compared against surface-based or satellite observations, with the co-location schemes matching data from the model grid to the real data.

The framework of maximising mutual information would be useful in the synthesis of multi-sensor retrievals over large spatial extents. With given time and length scales over which mutual information between data is high, data from multiple sources could be optimally combined to increase spatial or temporal coverage of satellite data, or to better characterise the uncertainties of satellite data via comparison with sufficiently nearby surface-based observations.

The co-location of data can also be extended to include more than two sources of data. Triple co-location (e.g McColl et al., 2014) is a method that already utilises three unique data sources to characterise the uncertainty and bias of retrievals with respect to the unknown true value. As long as a co-location scheme and homogenisation process can be defined that incorporates criteria combining more than two data sources, the framework can be utilised to identify the optimised choice of parametrisation for maximising the mutual information contained between the used data.

Maximising the mutual information between data is also essential for producing high quality labelled pairs of samples that can be used as the supervised training data for deep learning models. In order to produce accurate mappings from one data product to another, a deep machine learning approach not only requires high quality data, but a sufficient volume of samples to be trained on in order to learn the mapping. Generating paired data by maximising the mutual information produces data of high quality with limited contamination from independent data, whilst also ensuring that enough data is present for the structure in the joint distribution to be identified. The joint distribution structure is the probabilistic mapping from one data source to the other that is to be learned.

## 5   Summary

To summarise this work, we have proposed a data- and domain-agnostic framework that allows for the parameters determining the co-location of arbitrary data to be objectively optimised using the mutual information between the co-located data as an independent metric to assess the quality of the co-location. This is in opposition to using the validation or comparison metrics of subsequent analyses on the co-located data to assess the quality of the co-location, as is often done.

Correctly identifying the co-location parametrisation is crucial, as it determines the data available for all subsequent analysis and comparisons between the data. Parametrisations are multi-dimensional, and the effects of changing the parametrisation along individual dimensions are often non-separable. Thus sub-optimally selecting individual components of the parametrisation will degrade the subsequent analyses: either through the comparison of independent data or; by reducing the number of permitted dependent samples. Random or naive choices of the co-location parametrisation will almost certainly be sub-optimal, and the estimated mutual information surfaces are non-trivially dependent on the choice of parametrisation. We have shown that a one-size-fits-all approach to choosing the co-location parametrisation will likely be inappropriate when comparing data from different locations due to myriad local effects impacting the spatiotemporal variability of the geophysical fields being



measured, and that using the optimised co-location parametrisations we define yields better relationships between data to be compared than naive choices of co-location parametrisations.

We demonstrate the application of this framework by comparing ICESat-2 ATL09 vertically resolved cloud retrievals against Cloudnet retrievals from four observatories. We computed mutual information surfaces as a function of the co-location parametrisation $p$, and were able to identify site-specific optimised parametrisations $\hat{p}$ at each observatory. A basic comparison of ATL09 and Cloudnet VCF profiles for different parametrisations showed that, using our definition of optimised data co-location, comparisons between the data were improved over naive choices of co-location parametrisations, as well as a parametrisation typical of those used in the literature.

All that the framework requires in order to be implemented is: a co-location scheme with well defined criteria implementing the scheme, described by variable parametrisations $p$; a mutual information estimator that is appropriate for the data being compared and; a method for sampling different choices of $p$ in a way that the parametrisation that maximises the mutual information, $\hat{p}$, will be identified (grid-search or an implementation of gradient ascent). The framework is adaptable and widely applicable, with applications in satellite validations, satellite inter-comparisons, model validation, multi-sensor data synthesis and the production of labelled training data for deep learning methods.

*Code and data availability.* Software implementing the framework and producing results is given by Martin (2025a). The Cloudnet data used in this study are generated by the Aerosol, Clouds and Trace Gases Research Infrastructure (ACTRIS) and are available from the ACTRIS Data Centre using the following link: https://doi.org/10.60656/726097978e364d06. The specific dataset is given in Ebell et al. (2025). ICESat-2 ATL09 data are downloaded from the NASA Harmony API (https://harmony.earthdata.nasa.gov/) and utilise the ATL09 v6 data (Palm et al., 2023). Generated results can be accessed directly through Martin (2025b)

**Appendix A: Mutual information bounds for spatially inhomogeneous joint distributions**

One of the concepts underpinning the framework described in Sect. 2 is that the relationship between two retrievals depends on the spatial and temporal displacement between where the retrievals are made, and that a region within which an optimum comparison can be made exists. Geophysical fields are spatially inhomogeneous, so we can assume that the joint probability distribution relating two retrievals of geophysical fields is also spatially inhomogeneous.

Let us assume that for all displacements between the retrievals of two variables $X$ and $Y$, $r$, that the joint probability distribution relating $X$ and $Y$ can be described as $p(X,Y\,|\,r)$. This distribution will have some mutual information associated with it.

We will use the notation $\mathrm{I}(X;Y)$ to refer to the mutual information encoded between $X$ and $Y$, and $\mathrm{I}[\,p(X,Y)\,]$ to refer to the mutual information encoded by the probability distribution $p(X,Y)$. Thus, assuming that the joint probability $p(X,Y\,|\,r)$ is fixed, we can write

$$\mathrm{I}(X;Y|\boldsymbol{r}) = \mathrm{I}[\,p(X,Y|\boldsymbol{r})\,] = \mathrm{i}(\boldsymbol{r}). \tag{A1}$$





If we co-locate $X$ and $Y$ by sampling with a density $\lambda(\boldsymbol{r})$ across a domain $\mathcal{S}$, the joint probability distribution relating $X$ and $Y$ will be

$$p(X, Y \,|\, \mathcal{S}) = \frac{\int_{\mathcal{S}} d\boldsymbol{r} \, \lambda(\boldsymbol{r}) p(X, Y \,|\, \boldsymbol{r})}{\int_{\mathcal{S}} d\boldsymbol{r} \, \lambda(\boldsymbol{r})}, \tag{A2}$$

which represents a sample density weighted volume average of the probability distributions associated with all locations within the co-location domain $\mathcal{S}$.

Letting the integral of $\lambda(\boldsymbol{r})$ over $\mathcal{S}$ be $|\lambda_{\mathcal{S}}|$, the mutual information associated with the co-location within the domain $\mathcal{S}$ can be expressed as

$$\mathrm{I}(X; Y \,|\, \mathcal{S}) = \mathrm{I}\left[p(X, Y \,|\, \mathcal{S})\right], \tag{A3}$$

$$= \mathrm{I}\left[\int_{\mathcal{S}} \frac{d\boldsymbol{r} \, \lambda(\boldsymbol{r})}{|\lambda_{\mathcal{S}}|} p(X, Y \,|\, \boldsymbol{r})\right]. \tag{A4}$$

Mutual information can be expressed as the Kullback-Leibler (KL) divergence between a joint probability distribution and the product of its marginal distributions. It can also be shown that the KL divergence is convex in pairs of both of its arguments (e.g. Soch et al., 2025, proof 148). By extension, mutual information is convex in its arguments. This can be expressed through Jensen's inequality (Jensen, 1906):

$$\mathrm{I}[\kappa p + (1 - \kappa) q] \leq \kappa \mathrm{I}[p] + (1 - \kappa) \mathrm{I}[q], \tag{A5}$$

where $\kappa$ is a constant between $0$ and $1$ describing a mixture between probability distributions $p$ and $q$. The mutual information for the linear combination of probability distributions is less than or equal to the same linear combination of the mutual informations of the individual distributions. The inequality can be extended to a normalised weighted sum of multiple distributions. Thus, using Eq. (A4), we can express an inequality for $\mathrm{I}(X; Y \,|\, \mathcal{S})$ as

$$\mathrm{I}(X; Y \,|\, \mathcal{S}) = \mathrm{I}\left[\int_{\mathcal{S}} \frac{d\boldsymbol{r} \, \lambda(\boldsymbol{r})}{|\lambda_{\mathcal{S}}|} p(X, Y \,|\, \boldsymbol{r})\right], \tag{A6}$$

$$\leq \int_{\mathcal{S}} \frac{d\boldsymbol{r} \, \lambda(\boldsymbol{r})}{|\lambda_{\mathcal{S}}|} \mathrm{I}[p(X, Y \,|\, \boldsymbol{r})], \tag{A7}$$

$$\leq \int_{\mathcal{S}} \frac{d\boldsymbol{r} \, \lambda(\boldsymbol{r})}{|\lambda_{\mathcal{S}}|} \mathrm{I}(X; Y \,|\, \boldsymbol{r}) = \int_{\mathcal{S}} \frac{d\boldsymbol{r} \, \lambda(\boldsymbol{r}) \mathrm{i}(\boldsymbol{r})}{|\lambda_{\mathcal{S}}|}. \tag{A8}$$

Thus, if the mutual information between $X$ and $Y$ can be described by a function $\mathrm{i}(\boldsymbol{r})$ for samples recorded with a given displacement $\boldsymbol{r}$, the total mutual information when co-locating data within a domain $\mathcal{S}$ is bounded by the volume- and sample-density- weighted sum of all contributions.

For geophysical fields, we expect that their spatiotemporal autocorrelation (and by some extension, the mutual information) to be a decreasing function of the spatiotemporal displacements being considered. Thus, we can model $\mathrm{i}(\boldsymbol{r})$ as a decreasing function of $|\boldsymbol{r}|$. As such, it can be shown that the upper bound on the mutual information when considering all samples taken





within a $n$-spherical domain $\mathcal{S}(R)$ of radius $R$ around a fixed location is also a decreasing function of $R$. As stated in Sect. 2.4, this is the effect of data contamination by independent samples acting to decrease the mutual information encoded between the measurements $X$ and $Y$. The above analysis assumes that the computation of the mutual information is perfectly informed and not in fact data limited. As such, this is still consistent with our expectation that the maximum mutual information that will be evaluated for data co-located within a finite number of co-location events will not be found for $R = 0$, as the mutual information estimation will be data limited and thus the increase in $R$ will initially drive the value of $\hat{\mathrm{I}}(X;Y\,|\,\mathcal{S}(R))$ upwards towards the limiting value, until sufficient samples are available and the effects of contamination take over.

**A1   A radially isotropic two-population example**

Imagine a plane, with an observatory located at the origin, measuring variable $X$ with marginal distribution $p(X)$. There is also a mobile platform making point-like measurements of variable $Y$, with marginal distribution $p(Y)$. The measurements of variable $Y$ are made at distances $r$ from the origin, such that the sampling density is spatially uniform. This is represented as $\lambda(\boldsymbol{r}) = 1$ uniformly, and as such $\lambda$ will be ignored in the following derivation.

$Y$ is a spatially inhomogeneous variable, but isotropic (with respect to the origin), such that $X$ and $Y$ are related by a spatially varying joint probability distribution. For $r < R^*$, $X$ and $Y$ are dependent and have

$$p(X,Y\,|\,r < R^*) = p^*(X,Y), \tag{A9}$$

$$\mathrm{I}(X;Y\,|\,r < R^*) = \mathrm{I}[p^*(X,Y)] = \mathrm{I}^*, \tag{A10}$$

being related by the non-zero mutual information $\mathrm{I}^*$. Considering samples radially further away than $R^*$, $X$ and $Y$ are independent, such that

$$p(X,Y\,|\,r \geq R^*) = p(X)p(Y), \tag{A11}$$

$$\mathrm{I}(X;Y\,|\,r \geq R^*) = \mathrm{I}[p(X)p(Y)] = 0. \tag{A12}$$

The data co-location scheme for matching samples between $X$ and $Y$ is to consider all matches for which $r < R$. That is, the domain $\mathcal{S}$ is a disk of radius $R$ centred on the origin. As such, the joint probability distribution relating $X$ and $Y$ as a function of $R$ is

$$p(X,Y\,|\,r < R) = \begin{cases} p^*(X,Y) & R < R^*, \\ \left(\frac{R^*}{R}\right)^2 p^*(X,Y) + \left(1 - \left(\frac{R^*}{R}\right)^2\right) p(X)p(Y) & R \geq R^*. \end{cases} \tag{A13}$$

We can rewrite Eq. (A13) in terms of a mixing fraction $\kappa$,

$$p(X,Y\,|\,r < R) = \kappa p^*(X,Y) + (1-\kappa)p(X)p(Y), \tag{A14}$$

$$\kappa(R) = \begin{cases} 1, & R < R^*, \\ \left(\frac{R^*}{R}\right)^2, & R \geq R^*, \end{cases} \tag{A15}$$

$$0 < \kappa(R) \leq 1. \tag{A16}$$





Considering the convexity of the mutual information of mixture distributions, we can derive the bound on $I(X;Y\,|\,r<R)$:

$$I(X;Y\,|\,r<R) = I\left[\kappa p^*(X,Y) + (1-\kappa)p(X)p(Y)\right], \tag{A17}$$

$$\leq \kappa I\left[p^*(X,Y)\right] + (1-\kappa)I[p(X)p(Y)], \tag{A18}$$

$$\leq \kappa I(X;Y\,|\,r<R^*), \tag{A19}$$

$$\leq \kappa I^*. \tag{A20}$$

$$\therefore I(X;Y\,|\,r<R) \begin{cases} = I^*, & R < R^*, \\ \leq \left(\frac{R^*}{R}\right)^2 I^*, & R \geq R^*. \end{cases} \tag{A21}$$

Thus, for sufficiently large $R > R^*$, we necessarily expect a reduction in the mutual information between the co-located $X$ and $Y$ as the probability distribution becomes a mixture distribution, being contaminated with independent samples. Although this toy model is very simplified, it demonstrates explicitly how the inclusion of independent data reduces the upper bound on the mutual information encoded between $X$ and $Y$, and thought experiments can easily develop the model by including additional radial intervals within which the encoded mutual information within the interval is constant but between $I^*$ and $0$.

To demonstrate the mutual information bounds, we implemented the previously described sampling and co-location scheme. $X$ and $Y$ are two Gaussian distributed variables with unit variance. For $r < R^*$, $X$ and $Y$ have a bivariate Gaussian joint probability distribution, with correlation $\rho = \sqrt{1-e^{-2}}$, chosen such that $I^* = 1$ nat. For $r > R^*$, samples of $X$ and $Y$ are independently drawn from univariate Gaussian distributions to ensure an independent joint probability distribution between $X$ and $Y$.

Figure A1 shows the estimated $\hat{I}_{KSG}(X;Y)$ as a function of $1/N$, where $N$ is the number of samples of $X$ and $Y$ available to the KSG estimator. as $1/N \to 0$, the estimate $\hat{I}_{KSG} \to I$ should approach the real mutual information value. Figure A1 shows that for $N > 10^4$, the mutual information estimates for both the dependent and independent data have converged to the correct values of $1$ nats and $0$ nats respectively.

Figure A2 demonstrates a mixture distribution between the dependent and independent Gaussian joint distributions, as outlined in Eq. (A14–A16). $X$ and $Y$ consist of $N = 10^5$ joint samples, with $\kappa N$ samples being drawn from the dependent joint probability distribution, and $(1-\kappa)N$ samples being drawn independently. At the extremes where $\kappa = 0$ or $1$, we recover the results seen in Fig. A1, that the mutual information estimates agree with the actual mutual information values. However, for $0 < \kappa < 1$, we see the mutual information estimate is consistently lower than the theoretical upper bound provided by Eq. (A20).

Figure A3 extends the implementation used to create Fig. A2, by extending the definition of $N$ to depends on $R$, such that $N = \pi R^2 \lambda$. It also uses the full definition for $\kappa(R)$ given in Eq. (A15). The estimated profiles of $\hat{I}_{KSG}(X;Y\,|\,r<R)$ are plotted for multiple sampling densities, $\lambda$, defined by values $N^*$ such that

$$\lambda(N^*) = \frac{N^*}{\pi R^{*2}}. \tag{A22}$$

The mutual information profiles all follow a generic pattern. For $r < R^*$, increasing $r$ results in an increase in the number of dependent samples $N$ available for the KSG estimator to infer the relationship between $X$ and $Y$. This results in $\hat{I}_{KSG}$ rising





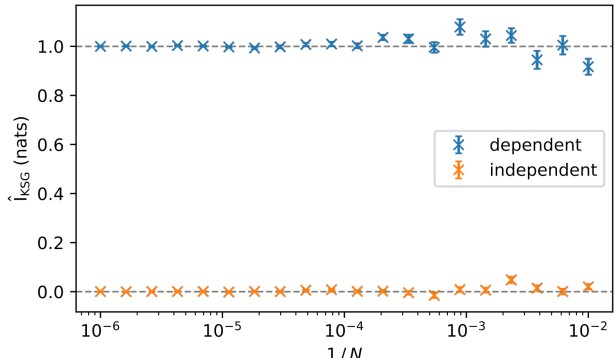

**Figure A1.** Mutual information estimates $\hat{I}_{KSG}$ between two Gaussian distributed variables as a function of the number of samples provided to the KSG estimator.

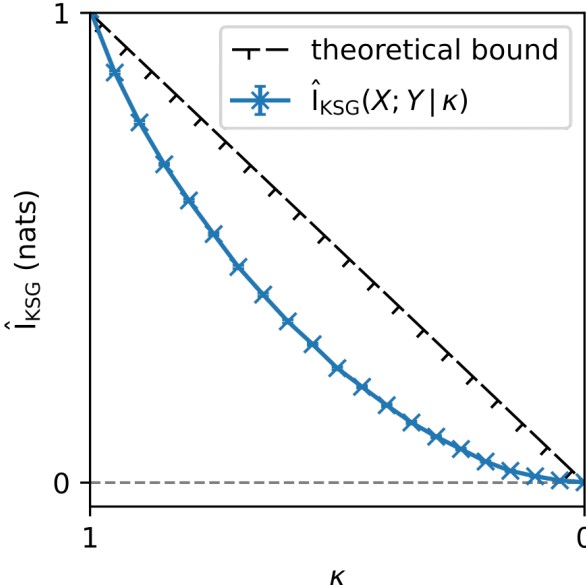

**Figure A2.** The mutual information between two variables $X$ and $Y$, which are sampled from a mixture distribution between a dependent and independent Gaussian distribution, as a function of mixing ratio $\kappa$. The theoretical upper bound given in Eq. (A20) is also plotted.

from $0$ nats when $r = 0$ towards a value of $I^*$. For $r > R^*$, the only additional samples provided to the KSG estimator are entirely independent, driving the estimates $\hat{I}_{KSG}$ towards $0$ nats. This is consistent with our expectations (see Sect. 2.4).

We also see, as in Fig. A2, that the mutual information estimates are almost always lower than the theoretical upper bound. The estimates for $r < R^*$ with $N^* = 10^2$ and $N^* = 10^3$ do vary above $I^*$ by more than $1\sigma_{KSG}$, but this can likely be attributed to the variance of the estimator with a low number of samples.





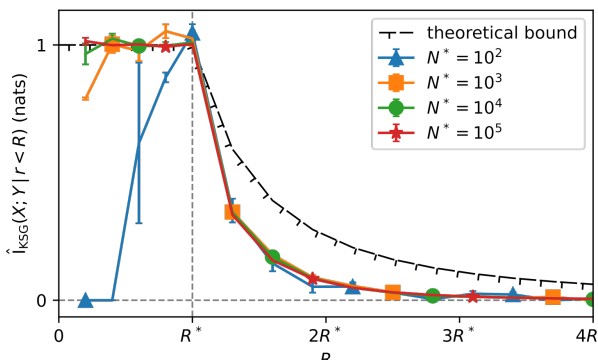

**Figure A3.** Mutual information estimates $\hat{I}_{KSG}(X;Y \,|\, r < R)$ plotted as functions of $R$ for different sampling densities $\lambda$, associated with different values $N^*$. The theoretical upper bound of the true mutual information between $X$ and $Y$ is also plotted, according to Eq. (A21).

## Appendix B: Quality checks and homogenisation

### B1 ATL09

The ICESat-2 ATL09 data (Palm et al., 2023) are downloaded through the NASA Harmony API (https://harmony.earthdata.nasa.gov/),

and are subset spatially based on circular polygons of radius $510$ km centred on each Cloudnet observatory. Code to facilitate the downloads through the NASA Harmony API is given in Martin (2025a).

Each individual ATL09 file can be associated with a unique co-location event. A given ATL09 file is opened and the *high_rate* group is loaded for each profile (one per ATLAS strong beam). The *high_rate* groups are each described by a temporal coordinate, with height and layer index coordinates associated with specific variables within the group.

For each temporal coordinate within a group, we create an empty boolean vector with the size of the vertical dimension. Elements of the vector are populated with true for elements corresponding to heights at which the ATL09 product reports cloud, as given in the *layer_bot* and *layer_top* variables. This forms a vertical cloud presence profile. The inclusion of a layer is rejected if the corresponding *layer_attr* value is not $1$ (cloud). If a layer within the vertical profile has an associated value *layer_conf_dens* $< 0.4$, then the entire vertical profile is rejected from the analysis.

All of the successfully generated vertical cloud presence profiles are concatenated. The profiles' *latitude* and *longitude* profiles are then used to compute the separation between the ATLAS footprint and the Cloudnet observatory, $r$. The co-location criteria outlined in Sect. 3.3 is applied.

If fewer than $17$ vertical cloud presence profiles remain after the co-location subsetting ($17 \times 240$ m $= 4.08$ km along-track distance), the co-location event is rejected for containing insufficient valid data. Otherwise, the VCF profile for the co-location

event is computed as the average vertical cloud presence profile at each height, considering all of the profiles permitted after the quality checks and co-location subsetting.



## B2 Cloudnet

The Cloudnet *categorize* data product is downloaded from the Cloudnet FMI website (https://cloudnet.fmi.fi), subset temporally between 01 October 2018 and 01 January 2025 to match the availability of ICESat-2 data (Ebell et al., 2025). Because the Cloudnet data is near-continuous compared to the ATL09 data at a given Cloudnet observatory, the ATL09 data is first processed to identify viable co-location events.

For a given successfully co-located ATL09 file, the time of closest approach is identified as the time $t_0$ for which the computed separation between the ATLAS footprints and the Cloudnet observatory, $r$, is minimised:

$$t_0 = t_j \,|\, j = \arg\min_k(r_k), \tag{B1}$$

where the subscript $k$ indexes the ATL09 vertical profiles loaded when co-locating the ATL09 data.

The interval

$$\mathcal{T} = \left[t_0 - \frac{\tau}{2}, t_0 + \frac{\tau}{2}\right] \tag{B2}$$

is defined, and all Cloudnet files with data falling within the temporal interval $\mathcal{T}$ are loaded and concatenated. The loaded data is then subset based on the interval $\mathcal{T}$. Co-location and quality subsetting can be considered as set intersection operations, which are commutative, so the data can be temporally subset prior to other operations being performed. This saves computational effort.

The *category_bits* variable is unpacked, and the cloudmask from the Cloudnet data is identified (according to the code from Tukiainen et al., 2020, defined in *cloudnetpy.products.classification._find_cloud_mask*) as

$$\text{cloud} = \text{droplet} \cup (\text{falling} \cap \text{freezing}), \tag{B3}$$

where droplet, falling and freezing are three of the unpacked boolean fields. The cloud variable is thus a boolean field with the *time* and *height* dimensions associated with the Cloudnet dataset. VCF profiles are computed as the temporal average of the cloud field across all profiles permitted by the temporal co-location.

## Appendix C: Across-track orbital density

In this appendix, we will derive a formula to determine the across-track density of satellite orbits, under the assumptions that:

1. The Earth is a sphere.

2. Subsequent orbits of the same satellite on the tangent plane at a given location on the Earth's surface are parallel.

3. Orbits form great circle paths over the surface of the Earth.





These assumptions are broken, but in most circumstances will lead to reasonable results. Firstly, the Earth is in fact not a sphere, but instead is better approximated as an oblate spheroid. However, the semi-major and semi-minor axes for the Earth
differ by less than $0.4\%$, making the spherical assumption reasonable for back-of-the-envelope calculations.

Secondly, treating subsequent orbits of a satellite on the tangent plane is a broken assumption. All of the orbits at a given latitude will have the same inclination relative to the vector locally pointing north. On the tangent plane, it is assumed that the direction pointing north is equal throughout the plane, however it will in reality have a longitudinal dependence. Sufficiently far from the poles, this assumption will be accurate to first order for displacements along the tangent plane, distances which
increase closer to the equator.

Finally, treating orbits as great circles is approximate, as the Earth rotates under the satellite as it orbits. This acts to make orbital tracks along the ground have a stronger westwards component than they otherwise would, effectively tilting the orbital track relative to a co-rotating great circle of the same inclination. For polar orbiting satellites with orbital inclinations greater than 90 °, this acts to make the orbital track locally appear to have a shallower angle relative to the equator compared to the
great circle with the same inclination.

We can define three points on the surface of the Earth: let $A_0$ be the ascending node of the orbit, found on the equator; let $A_1$ be the location on the Earth's surface at which we want to compute the local across-track orbital density and; let $A_2$ be the point where the highest latitude is reached. If we let $\phi$ represent latitude (positive northwards), and $\lambda$ be longitude, the locations can be expressed as

$$(\phi, \lambda) \tag{C1}$$

$$A_0 : (0, -\lambda_0), \tag{C2}$$

$$A_1 : (\phi_1, 0), \tag{C3}$$

$$A_2 : (\phi_2, \lambda_2), \tag{C4}$$

where $\phi_1$ and $\phi_2$ are known constants, and we have the constraint $\lambda_0 + \lambda_2 = \frac{\pi}{2}$. Specifically, $\phi_2$ is equal to the reference angle
of the orbital inclination of the satellite (the angle as it would be if limited to being between 0 and $\frac{\pi}{2}$).

We will describe the bearing along which the great circle from point $a$ to point $b$ lies when measured from point $a$ as $\alpha_{ab}$. The angle $\alpha_{ab}$ can generally be expressed as

$$\tan \alpha_{ab} = \pm \left( \frac{\cos \phi_b \sin \lambda_{ab}}{\cos \phi_a \sin \phi_b - \sin \phi_a \cos \phi_b \cos \lambda_{ab}} \right), \tag{C5}$$

where $\lambda_{ab} = \lambda_b - \lambda a$ is the difference in longitude between point $a$ and point $b$.
At $A_2$, the orbit has reached its highest latitude, $\phi_2$. As such, the bearing of the orbit must necessarily be $\frac{\pi}{2}$ as the satellite transitions from heading northwards to southwards. Thus, $\alpha_{20} = \alpha_{21} = \frac{\pi}{2}$. Knowing that $\lim_{\left(\alpha \to \frac{\pi}{2}\right)} \tan \alpha \to \pm \inf$, we know





that the denominator in Eq. C5 must tend towards zero. For the case of $\alpha_{20}$:

$$\cos\phi_2\sin\phi_0 - \sin\phi_2\cos\phi_0\cos\lambda_{20} = 0 \tag{C6}$$

$$\implies \sin\phi_2\cos\lambda_{20} = 0 \tag{C7}$$

$$\implies \cos\lambda_{20} = 0 \tag{C8}$$

$$\implies \lambda_{20} = \frac{\pi}{2}. \tag{C9}$$

This agrees with our previous constraint about the longitudinal separation between $A_0$ and $A_2$.

The same logic can be applied for $\alpha_{21}$, deriving an equation for $\lambda_{12} = \lambda_2$:

$$\cos\phi_2\sin\phi_1 - \sin\phi_2\cos\phi_1\cos\lambda_{12} = 0 \tag{C10}$$

$$\cos\lambda_{12} = \frac{-\cos\phi_2\sin\phi_1}{\sin\phi_2\cos\phi_1} \tag{C11}$$

$$= -\left(\frac{\tan\phi_1}{\tan\phi_2}\right). \tag{C12}$$

Now, applying Eq. (C5) and Eq. (C12) to finding $\alpha_{12}$, we get

$$\tan\alpha_{12} = \left(\frac{\cos\phi_2\sqrt{1-\cos^2\lambda_{12}}}{\cos\phi_1\sin\phi_2 - \sin\phi_1\cos\phi_2\cos\lambda_{12}}\right), \tag{C13}$$

$$= \left(\frac{\cos\phi_2\sin\lambda_{12}}{\cos\phi_1\sin\phi_2 - \sin\phi_1\cos\phi_2\cos\lambda_{12}}\right), \tag{C14}$$

$$\vdots \tag{C15}$$

$$\tan\alpha_{12} = \frac{\cos\phi_1\cos^2\phi_2\left(\tan^2\phi_2 - \tan^2\phi_1\right)^{\frac{1}{2}}}{\cos^2\phi_1\sin^2\phi_2 + \sin^2\phi_1\cos^2\phi_2}, \tag{C16}$$

which is importantly expressed purely as a function of the known latitudes $\phi_1$ and $\phi_2$.

In order to convert this bearing in to an across-track density of orbits, we need to evaluate the perpendicular separation between adjacent orbital tracks. For a satellite that revisits the same along-ground orbital track every $N$ orbits, the longitudinal displacement between adjacent tracks, $\delta x$, is given as

$$\delta x = \frac{R\cos\phi_1}{N}. \tag{C17}$$

This can be related to the across-track separation between the adjacent orbits, $\delta s$, as

$$\frac{\delta s}{\delta x} = \cos\alpha_{12}. \tag{C18}$$





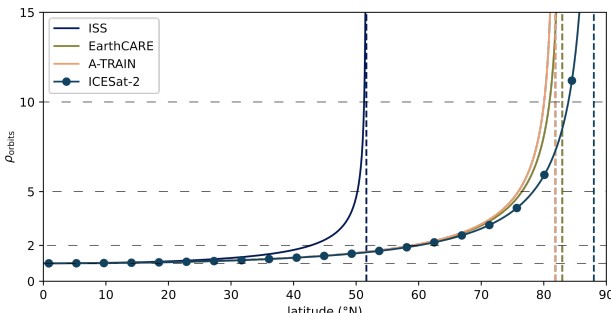

**Figure C1.** Normalised across-track density of orbits, $\rho_{\mathrm{orbits}}$, for the ISS, EarthCARE, A-Train satellite constellation, and ICESat-2. Horizontal dashed lines have fixed densities of 1, 2, 5 and 10. Vertical dashed lines correspond to the highest latitudes achieved by each satellite, and represent where Eq. (C21) diverges.

The across-track density of orbits, $\rho_{\mathrm{orbits}}$ is inversely proportional to $\delta s$, such that

$$\rho_{\mathrm{orbits}} \propto \frac{1}{\delta s}, \tag{C19}$$

$$\propto \frac{N}{R \cos \alpha_{12} \cos \phi_1}, \tag{C20}$$

$$\propto \left( \frac{(\cos \phi_1)^{-1}}{\cos \left( \arctan \left( \frac{\cos \phi_1 \cos^2 \phi_2 (\tan^2 \phi_2 - \tan^2 \phi_1)^{\frac{1}{2}}}{\cos^2 \phi_1 \sin^2 \phi_2 + \sin^2 \phi_1 \cos^2 \phi_2} \right) \right)} \right). \tag{C21}$$

Thus, the across-track density of orbits can be approximated given the reference angle of the satellite's orbital inclination, $\phi_2$, and the latitude of the location at which the density is to be calculated, $\phi_1$. Multiplying Eq. (C21) by a factor of $\sin \phi_2$ normalises $\rho_{\mathrm{orbits}}$ such that $\rho_{\mathrm{orbits}} = 1$ when $\phi_1 = 0$.

Figure C1 shows the across-track density of orbits calculated for the ISS, EarthCARE, the A-Train constellation of satellites and ICESat-2.

*Author contributions.* ASM conceptualised the study; ASM designed the framework; ASM, HG, MG designed the analysis methodology; ASM wrote the analysis software; ASM performed the analysis; All co-authors analysed and reviewed the results; ASM prepared the manuscript with feedback and contributions from all co-authors; HG, RN provided supervision.

*Competing interests.* The contact author has declared that none of the authors has any competing interests.



*Acknowledgements.* This work was supported by a SENSE studentship to ASM [2748913], funded by the Natural Environment Research Council [NE/T00939X/1]. We acknowledge ACTRIS and Finnish Meteorological Institute for providing the Cloudnet data set which is available for download from https://cloudnet.fmi.fi. The cloud radar data for Ny-Ålesund was provided by the University of Cologne, the ceilometer and microwave radiometer data by the Alfred Wegener Institute, Helmholtz Centre for Polar and Marine Research. We thank the staff of AWIPEV research base in Ny-Ålesund for technical support of the measurements. We gratefully acknowledge the funding by the Deutsche Forschungsgemeinschaft DFG (German Research Foundation) - project number 268020496 - TRR 172, within the "Transregional Collaborative Research Center 'ArctiC Amplification: Climate Relevant Atmospheric and SurfaCe Processes, and Feedback Mechanisms (AC)3'". We acknowledge ECMWF for providing IFS model data, DWD for providing ICON model data, and NCEP (National Centers for Environmental Prediction) for providing access to GDAS1 data. The authors would like to thank the many teams contributing to maintaining ICESat-2 for their ongoing efforts in creating the atmospheric data products. This work used JASMIN, the UK's collaborative data analysis environment (https://www.jasmin.ac.uk; Lawrence et al., 2013). The Scientific colour maps acton, hawaii, imola, lipari, navia, roma and vik (Crameri, 2023) are used in this study to prevent visual distortion of the data and exclusion of readers with colour-vision deficiencies (Crameri et al., 2020). ASM would like to thank Von P. Walden for contributions to the statistical methodology, and for his shared guidance and wisdom. ASM would like to acknowledge the rest of the ICECAPS team and their support over the past two years. ASM would like to thank WO, SH, ISSW and TM for making the long two weeks enjoyable, and KC for always being there when the going gets tough.



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
