# Peer review of "A guide to optimised spatiotemporal data co-location by mutual information maximisation"

_EGUsphere, 2025_

## Referee Comment (RC2)

**Referee Report on "A guide to optimised spatiotemporal data co-location by mutual information maximisation"**

**Dear Authors,**

**your manuscript is very well motivated and clearly written. It addresses a well-known limitation of traditional co-location approaches that rely on fixed spatial and temporal thresholds. The use of information-theoretic metrics provides a principled and potentially powerful alternative. However, several aspects of the methodology, assumptions, and practical applicability would benefit from further clarification, quantitative justification, and discussion before the paper is ready for the publication.**

General Comments:

1. The Introduction is very informative but I would consider it too long. I would recommend to make it a bit more compact while keeping all relevant references in.
2. The paper is lengthy, so I would recommend to try out ways to reduce its length as it could be a big discouraging for a potential reader to read it, even though its topic of discussion is of a great significance. Towards this aim, I would recommend to move some from the Appendices to a Supplement, which will come in a separate document. To my opinion, Appendix B is a good candidate to be moved to the Supplement. Moreover, Appendix C could become another section in the supplement. If you still decide to keep it in the manuscript, please try to make it shorter. I believe that Appendix A should still stay in the manuscript.
3. The section 4 "Discussion" is also too lengthy. I would not create separate subsections for 4.1, 4.2 and 4.3. I would just be merging it to one while explaining the important parts of the frameworks "The choice of mutual information estimator", "Physical interpretation of $\hat{p}$" and "Choice of co-location scheme".
4. Section 3.6.3 contains some weak statements as the comparison of the two extreme parameterizations $p_{00}$ and $p_{11}$ is not that strong. Try to make this section compact and more meaningful.
5. I would recommend to completely remove Section 3.7 too. I do not clearly see any strong point to keep it. On the contrary, the manuscript will benefit from the change as it will become shorter.
6. The Figures have some issues with the axis labels. The font size is way too small. When the reader prints the manuscript, the axes are not even readable. Even on a typical-size laptop screen, the axes are not visible and only with zooming in you can read them well. But this interrupts the flow of the reading. I would recommend to simply increase the font size of all axes in the figures.

Specific Comments:

1. The reference (Loew et al, 2017) appears too frequent in the Introduction; precisely four times and then again in the second sentence of the Section 2 "Framework". It seems like a repetition.
2. Line 60 – I would not start the sentence "This should be avoided". Please rephrase it.
3. Introduce numbering items for the co-location scheme, criteria, parameterization and event. Their definition is good as it introduces early enough the terminology to the reader. It is very much appreciated.

4. Figure 1 is very good introductory of the potential schemes. Please add main references for each scheme to the figure legend. There are many references in the text but it is still nice to have the visualization of the scheme together with the main reference(s).

5. Line 105 – What "best results" mean? Earlier in the introduction (line 60), it was recommended that this should be avoided.

6. Line 122-124 – The sentence "For example, … ancillary wind data" is disconnected with the previous text. It only makes sense with prior explanation why/how local advection might affect the colocation scheme. It should be further explained and ideally be moved to another location in the text.

7. Figure 3 is very nice. Please make clearer that the preferred situation is depicted in plot 3b.

8. Line 221 – Remove the first bracket from Eq. 3. It should appear as

$$\hat{p} = \arg\max_{p \in \mathcal{P}} \hat{I}(p)$$

9. Line 233 – The vertical (range) resolution is on the order of decimeters ($\sim$10–30 cm), determined by timing precision. 30 m should refer to the horizontal segment length, not the vertical resolution. Do you mean the following: "Photon returns are accumulated into 30 m along-track segments, which are further aggregated over 400 consecutive laser pulses to produce a product with an effective along-track resolution of approximately 240 m."?

10. Line 278 – There are some problems with Eq.5: (a) The vertical bar | is ambiguous here (it usually means "such that" in set notation or conditional probability), (b) The prime notation j' is unnecessary and makes it harder to read. The easier is to express it like

$$j' = \arg\min_{j} r_j, \qquad t_0 = t_{j'}.$$

11. Line 286-287 – "The observed clouds are optically thick enough to attenuate the ATLAS lidar beam throughout the co-location event, so lower-level cloud layers will be missed.". This is an important information which can have an important effect on the comparisons against the ground-based VCF. However, I would expect that the opposite might happen for the Cloudnet data. The higher-level cloud layers will be then missed? Or what is the situation exactly?

12. Figure 4 – There are some issues with those plots, apart from the general comment that fonts are tiny and the several plots are not well read or displayed. Fig 4c should indicate a minimum of 18.4km, if I understood well? Please add the information to the plot. Also, it might be preferable to show the area which corresponds to the contributing pairs ATL09-Cloudnet at the co-location parameterization. Fig. 4f shows that the profiles have huge differences for altitude 4km and below. How do the uncertainties look like for the two measurements? Are the uncertainties taken into consideration at all? It would be highly advisable to include the error bars into the VCF profiles. Moreover, it seems like there is an overlap issue in the Cloudnet profile. Can you please explain that? The VCF profile increases exponentially from the ground up to the first 1.0-1.5km where it reaches the value of 1.0 (i.e., fully cloudy). At the ATL09 profile the VCF is exactly zero, which could imply that the ATLAS laser beam is completely attenuated at those altitudes. But then, the vertical bins should be removed from the dataset in a pre-processing step. Is this performed? Can you also elaborate on the potential problem of incomplete overlap of the Cloudnet lidar? The incomplete overlap problem refers to the situation in which, at short ranges from a lidar system, the laser beam and the telescope field of view (FOV) do not fully overlap. Remaining incomplete overlap problems in the retrieved vertical profiles could introduce artifacts which should be removed in a kind of pre-processing step. Could you please include some more

VCF profiles for this Cloudnet station, as well as for the other three Cloudnet stations to clarify the situation?

13. Line 301-302 – "Above 5 km in height, both VCF profiles visually correlate with each other, indicating that the co-location ... unobserved by ICESat-2.": Could it be that simply above 5 km, the ground-based lidar face laser beam attenuation issues? Or this is certainly not a case in such conditions of optically thick clouds present?

14. Line 311 – Can you elaborate more on the selection of k=10?

15. Line 319-320 – Further, can you elaborate on the $n_i=10$ and $n_{repeats}=20$ selection?

16. Line 325 – And if null hypothesis is rejected, then p is not a good candidate, right? Probably refer to the t-test statistical test you make. Consider rephrasing to "*For each* p≠p^p \neq \hat{p}p=p^, *we apply a Welch's t-test to evaluate the null hypothesis that the mean estimated mutual information* I^KSG(p)\hat{I}_{\mathrm{KSG}}(p)I^KSG(p) *does not differ significantly from* I^KSG(p^)\hat{I}_{\mathrm{KSG}}(\hat{p})I^KSG(p^). *Parametrizations for which the null hypothesis is not rejected at the 0.05 significance level are considered candidate optimized parametrizations.*".

17. Line 331 – Could you explain how the measurement uncertainties are counted into the selection of partially cloudy category of very small VCF (e.g. VCF < 0.1)?

18. Line 341 – Please revise Eq. 8. Option A (copula in terms of uniform variables):
$$C(u,v) = P(U \leq u, V \leq v)$$
Option B (copula expressed with the original variables):
$$C(F_X(x), F_Y(y)) = P(X \leq x, Y \leq y)$$
Both are correct; which one you use depends on context and clarity.

19. Figure 5 – Plot 5c looks like there is contamination from independent data? It appears very similar to Figure 3d. Please clarify this point. Also, if there are no unique parameterization and the hatching shows several possible good parameterization candidates, then some analysis should be included which shows and explains how much the impact could be.

20. Line 397 – "As with $N_{events}$, the smooth color gradient ... and τ." But, $N_{events}$ have not been expressed as a function of τ. Please clarify this discrepancy.

21. Line 418-419 – I am not sure if I interpreted correctly. Do you imply that the difference of 10 hours (form 8 hours to 18 hours) is significant or not?

22. Figure 6c – The dispersion of the possible optimal co-location parameterization is high. There is some explanation/interpretation in the text which associates this to the flat terrain around the ground-based station. Could you give an estimate of how much such a condition could impact the comparisons over Hyytiala?

23. Line 427-428 – It appears like there is contamination from independent samples like in Fig. 3d. Could you please clarify this point and probably rephrase the sentence "regions where the input data to the estimator is contaminated with independent samples"?

24. Line 434 – The statement "are similar orders of magnitude" is a bit misleading here as the minimum of 4 hours is less than half the maximum of 10 hours. Please rephrase that.

25. Line 435 – There is a statement that the temporal values τ are consistent with other studies but there is no statement about the spatial values R. Are also those consistent with available literature?

26. Line 455 – If I understood correctly, the selection of 100-dimensional joint probability distribution constrains the sampling in the other 3 locations. Probably for each location, depending on the

latitude that the ground station lies, other selection should be done? What is the reasoning of having 100 dimensions?

27. Line 476-480 – I cannot clearly see what it is hidden in those sentences. Could you proof that another set of co-location parameterization would not be suitable? Could you think of any validation of the optimal co-location parameterization?

28. Figure 7 – Instead of comparing with the two extremes $p_{00}$ and $p_{11}$, I would also compare the confusion matrices for other potential candidates of a co-location parameterization. Probably, I would pick the one with the smallest temporal window and the one with the largest spatial extend. Please also add in the legend what $p_{00}$ and $p_{11}$ mean. The parameterization is given in the text but the plots should also be self-informative without the need to read all the text. I also have a problem with the group of plots e-h: can they be biased due to the first 1.0-1.5 km that the VCF exponentially increases with height (w.r.t/ the incomplete overlap)? Please provide some more cases including also the other Cloudnet stations. If the VCF profiles have the same structure, then consider to apply some filtering/pre-processing prior to the co-location parameterization selection scheme.

29. Line 504-505 – From the statement I understand that $p_{lit}$ is still a good choice. Therefore, the empirical choice of the parameters $\tau$ and $R$ as taken from the literature appear as good as the one that you found from the maximum mutual information. Please clarify this point.

30. Line 508-509 – "The shapes of … at lower values." Could that be due to problems of incomplete overlap? Please clarify this point.

31. Figure 8 – There are essential problems with the visibility of the axis's labels in printed form. Please try to make them also friendlier for people with color vision deficiencies (CVD); certain color combinations are hard or impossible to distinguish and should be avoided.

32. Line 528-530 – I would not conclude the session with such a strong sentence. There is no validation which proves the statement, right?

33. Line 534-535 – Could this be related to signal attenuation from the ground-based lidar? Please comment on this.

34. Line 824 – Equation B1 could be rephrased similarly to Eq.5 at Line 278

35. Line 880 – Equations C10-C11 must have a wrong sign. Please check if the minus sign (-) must be present in the aforementioned equations.

36. Figure C1 – Why do you include in the plot also the ISS, EarthCARE and A-Train satellites? Relevant to that, there is some text in lines 901-902 about those satellites. Please provide a statement of why would this be relevant to this manuscript? Do you foreseen to apply this co-location parameterization to other missions?

Technical Corrections:

37. Line 16 – Typo of "validation an constraining" should be "validation and constraining".
38. Line 62 – Remove the second "to the".
39. Line 103 – Remove the second "be"
40. Line 118 – "Co-location" with capital letter.
41. Line 229 – Please correct the ATLAS acronym to Advanced Topographic Laser Altimeter System (ATLAS)
42. Line 452 – There is an extra "a". Please remove it.

43. Line 541-543 – This sentence reads like being replicated to the lines 535-537. Please remove or rephrase.
44. Line 562-563 – In case you do not remove the complete section 3.7, please remove the conclusive sentence.
45. Line 776 – "As 1/N .." with capital A.
46. Line 784 – The sentence does not read well. Remove either the word "is" or add the word "that" after "we see".
47. The doi of the reference "Herzfeld, U., Palm, S., and Hancock, D.: Ice, Cloud, and Land Elevation Satellite (ICESat-2) Project Algorithm Theoretical Basis Document for the Atmosphere, Part 2: Level 2 and 3 Data Products, version 4, https://doi.org/10.5067/JT3IX3YFW1RR, publisher: NASA National 960 Snow and Ice Data Center Distributed Active Archive Center, 2021b." is wrong.
48. The same for the doi of the reference "Palm, S., Yang, Y., Hertzfeld, U., and Hancock, D.: Ice, Cloud, and Land Elevation Satellite (ICESat-2) Project Algorithm Theoretical Basis Document for the Atmosphere, Part I: Level 2 and 3 Data Products, version 4, https://doi.org/10.5067/VZIDW16UA4S1, publisher: NASA National Snow and Ice Data Center Distributed Active Archive Center, 2021a."
49. The same for the doi of the reference "Vinjamuri, K. S., Vountas, M., Lelli, L., Stengel, M., Shupe, M. D., Ebell, K., and Burrows, J. P.: Validation of the Cloud_CCI (Cloud Climate Change Initiative) cloud products in the Arctic, Atmospheric Measurement Techniques, 16, 2903–2918, https://doi.org/10.5194/amt-16- 1095 2903-2023, publisher: Copernicus GmbH, 2023."